# RadImageNet-VQA: A Large-Scale CT and MRI Dataset for Radiologic Visual Question Answering

**Léo Butsanets**[1]                               LEO.BUTSANETS@RAIDIUM.EU
**Charles Corbière**[1]                   CHARLES.CORBIERE@RAIDIUM.EU
**Julien Khlaut**[1,2,3]                       JULIEN.KHLAUT@RAIDIUM.EU
**Pierre Manceron**[1]                  PIERRE.MANCERON@RAIDIUM.EU
**Corentin Dancette**[1]                CORENTIN.DANCETTE@RAIDIUM.EU

[1]*Raidium, Paris, France.*

[2]*Université de Paris Cité, Hôpital Européen Georges Pompidou, AP-HP, Paris, France.*

[3]*Department of Vascular and Oncological Interventional Radiology, INSERM, Paris, France.*

**Editors:** Accepted for publication at MIDL 2026

## Abstract

In this work, we introduce *RadImageNet-VQA*, a large-scale dataset designed to advance radiologic visual question answering (VQA) on CT and MRI exams. While existing medical VQA datasets are limited in scale, dominated by X-ray imaging or biomedical illustrations, and prone to text-based shortcuts, RadImageNet-VQA is built from expert-curated annotations and provides 750K images paired with 7.5M QA samples. It covers three key tasks—abnormality detection, anatomy recognition, and pathology identification—spanning 8 anatomical regions and 97 pathology categories, and supports open-ended, closed-ended, and multiple-choice questions. Extensive experiments show that state-of-the-art vision-language models still struggle with fine-grained pathology identification, especially in open-ended settings and even after fine-tuning. Text-only analysis further reveals that model accuracies collapse to near-random without image inputs, confirming that RadImageNet-VQA is free from linguistic shortcuts. The full dataset and benchmark are publicly available at https://huggingface.co/datasets/raidium/RadImageNet-VQA.

**Keywords:** Medical Visual Question Answering, Vision-Language Models, Radiology.

## 1. Introduction

Interpreting radiology exams requires integrating complex visual patterns with clinical reasoning, often with medical context (Yildirim et al., 2024; Dancette et al., 2025; Khlaut et al., 2025). Vision-language models (VLMs) (Li et al., 2023b; Bai et al., 2025; Chen et al., 2024; Liu et al., 2023) have recently achieved strong performance on general multimodal benchmarks. Medical variants, such as MedGemma (Sellergren et al., 2025) and Lingshu (Xu et al., 2025b), now incorporate curated medical data to better align with clinical knowledge. These models are particularly attractive for radiology, where generating reports and supporting clinical reasoning require combining visual and textual information. Yet, evaluating free-text radiology reports remains difficult: commonly used similarity metrics often fail to capture clinical correctness, factual consistency, or semantic alignment with ground-truth findings (Irvin et al., 2019; Delbrouck et al., 2024; Xu et al., 2025a). In contrast, radiologic visual question answering (VQA) offers a more structured alternative to probe

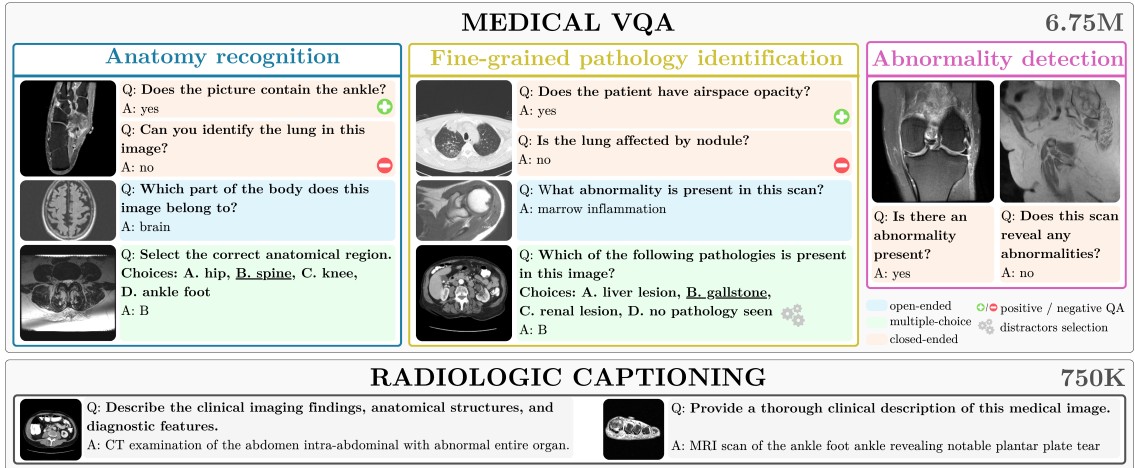

Figure 1: **Overview of the RadImageNet-VQA dataset**, which provides radiology-focused supervision across three VQA tasks (anatomy recognition, fine-grained pathology identification, and abnormality detection) using diverse open-ended, closed-ended, and multiple-choice formats. It also includes radiologic captioning pairs for image–text alignment.

model reasoning. Its question-answer format mirrors how radiologists interrogate imaging findings, providing an interpretable way to evaluate a model's image-grounded reasoning.

Existing medical VQA datasets (Lau et al., 2018; Liu et al., 2021; Pal et al., 2025; Zhang et al., 2023) present several limitations that hinder proper evaluation of these models. Early efforts rely on limited expert annotation and thus offer small collections with narrow anatomical and pathological coverage (Lau et al., 2018; Liu et al., 2021). Larger recent datasets draw heavily from X-ray images or biomedical figures scraped from publications (Pal et al., 2025; Zhang et al., 2023), providing little representation of other common imaging modalities, in particular CT or MRI. Many also contain textual shortcuts that allow models to answer correctly without interpreting the image, as shown in Section 4.2. Beyond evaluation, progress in medical VLMs also requires access to large, well-curated training data aligned with diagnostic tasks. This emphasizes the need for large-scale VQA resources that spans multiple radiologic tasks while minimizing possible shortcut cues to offer a more rigorous foundation for assessing and advancing medical VLMs.

In this paper, we introduce *RadImageNet-VQA*, a large-scale dataset designed for training and benchmarking radiologic VQA on CT and MRI exams (see Figure 1). Built from the CT/MRI subset of RadImageNet (Mei et al., 2022) and its expert-curated anatomical and pathological annotations, RadImageNet-VQA provides 750K images with 7.5M generated samples, including 750K medical captions for visual-text alignment and 6.75M question-answer pairs that span three radiology tasks: fine-grained pathology identification, anatomy recognition, and abnormality detection. The dataset includes open-ended, closed-ended, and multiple-choice questions across 8 anatomical regions and 97 pathologies, where questions were generated with prompt-based templates and constructed to probe visual-grounded understanding while minimizing text-only shortcut answering. For evaluation, we construct a stratified benchmark of 1,000 images with 9,000 question-answer pairs covering

all tasks and question types. Through extensive zero-shot evaluation and fine-tuning experiments with RadImageNet-VQA, we show that while anatomy recognition is nearly solved for current VLMs, pathology identification remains a major bottleneck, especially for open-ended responses. Text-only analysis further confirm that RadImageNet-VQA greatly reduces shortcut cues present in existing datasets. Fine-tuning yields substantial gains across models, yet we observe that initializing models with medically pretrained vision encoders such as MedSigLIP (Sellergren et al., 2025) provides no advantage over general-purpose encoders.

Our main contributions are as follows:

• We release RadImageNet-VQA, a radiologic dataset that includes both a curated benchmark and a large training corpus. It spans 8 anatomical regions, 97 pathologies, and three task families (abnormality, anatomy, pathology) with multiple question formats, providing 750K images paired with 7.5M generated samples.

• We conduct an extensive evaluation of state-of-the-art VLMs showing that existing models perform well on anatomy and basic abnormality detection but still struggle with fine-grained pathology identification. Text-only ablations confirm that RadImageNet-VQA minimizes shortcut cues.

• Fine-tuning on RadImageNet-VQA produces substantial gains across all model families, while medical pretrained vision encoders do not improve downstream performance.

## 2. Related Work

### 2.1. Vision-language models in radiology

Recent VLMs, such as LLaVA (Liu et al., 2023), QwenVL (Bai et al., 2025), InternVL (Chen et al., 2024), typically combine a vision encoder, often initialized from CLIP (Radford et al., 2021) or SigLIP (Zhai et al., 2023) backbones, with a large language model (LLM) via a lightweight adapter module. In the medical domain, progress has largely been driven by data-centric adaptation, where general-purpose VLMs are aligned to medical images through image–text pretraining and multimodal instruction tuning (Li et al., 2023a; Sellergren et al., 2025; Xu et al., 2025b). While several 3D radiology VLMs have recently been proposed (Blankemeier et al., 2024; Wu et al., 2023; Xin et al., 2025; Ates et al., 2025), their development is constrained by the scarcity of large volumetric datasets, making broad and diverse 2D datasets still essential for training and evaluation. Beyond training, reliable evaluation also remains challenging. Radiology report generation (Bannur et al., 2024) is appealing but difficult to assess automatically, as standard metrics correlate poorly with clinical correctness. On the other hand, visual-question answering (VQA) provides a more controlled and interpretable proxy framework for probing clinical reasoning grounded in images. This has motivated increasing interest in radiologic VQA benchmarks, particularly for CT and MRI, which remain underrepresented in existing resources.

### 2.2. Medical VQA datasets

Early radiologic VQA, such as VQA-RAD (Lau et al., 2018) and SLAKE (Liu et al., 2021) introduced structured QA annotations but are limited by scale and by narrow anatomical and pathological coverage. More recent efforts, such as the medical subset of MMMU (Yue

| Dataset | Modality | Question Types | # Images | # QAs Train | # QAs Test |
|---------|----------|----------------|----------|-------------|------------|
| VQA-RAD (Lau et al., 2018) | X-ray, CT | open, closed | 0.3K | 2.8K | 0.45K |
| SLAKE (Liu et al., 2021) | X-ray, CT, MRI | open, closed | 0.6K | 4.9K | 1.1K |
| MMMU-Med (Yue et al., 2024) | diverse | multiple-choice | 1.9K | − | 1.8K |
| ReXVQA (Pal et al., 2025) | X-ray | multiple-choice | 160K | 573K | 40.8K |
| M3D-VQA (Bai et al., 2024) | 3D CT | multiple-choice | 96K | 662K | 2K |
| 3D-RAD (Gai et al., 2025) | 3D CT | open, closed | 16K | 136K | 34K |
| **RadImageNet-VQA (Ours)** | **CT, MRI** | **open, closed, multiple-choice** | **750K** | **7.5M** | **9K** |

Table 1: **Comparison of radiologic VQA datasets** compared by imaging modality, supported question types, and number of images, and the number of question-answer pairs for training and evaluation. RadImageNet-VQA provides the largest CT/MRI coverage and the most extensive QA corpus.

et al., 2024) extend medical breadth, yet their radiology content is sparse and relies largely on images from textbooks or publications rather than CT or MRI. Programmatic datasets such as ReXVQA (Pal et al., 2025) improve scalability but are restricted to X-ray images and therefore cover only a small portion of radiology practice. Table 1 summarizes these datasets and highlights the narrow CT/MRI coverage of existing benchmarks. Additionally, a persistent issue across these benchmarks is the presence of linguistic shortcuts, label phrasing, or distributional biases that allow models to guess without relying on images In parallel, several 3D medical VQA benchmarks, such as DeepTumorVQA (Chen et al., 2025b), M3D-VQA (Bai et al., 2024), and 3D-RAD (Gai et al., 2025), extend to volumetric tasks. Although clinically meaningful, these datasets cover limited anatomy and pathologies, require extensive expert curation, and remain small in scale. Current 3D VLMs are trained on small volumetric corpora and lack the broad visual priors available to large 2D models. Strengthening 2D CT/MRI supervision therefore remains essential for building radiology VLMs, even as 3D approaches continue to mature.

## 3. RadImageNet-VQA Dataset

In this section, we describe the construction process of RadImageNet-VQA dataset, as illustrated in Figure 2, and detail the statistics and metrics of the benchmark subset.

### 3.1. Dataset construction pipeline

Data is sourced from RadImageNet (Mei et al., 2022), a large expert-annotated medical imaging dataset in which each image is associated with a modality (CT, MRI, US), a body part (e.g., abdomen, hip, brain) and a pathology label. From this resource, we use the CT and MRI subsets to form the basis for generating clinically meaningful captions and VQA samples across anatomy, abnormality, and fine-grained pathology tasks.

**Radiologic captioning generation.** VLMs are typically trained with an alignment stage in which the model learns to associate visual content with textual semantics using large collections of image–caption pairs. As presented in Figure 2, to support a clinically meaningful alignment dataset in training, we leverage the rich metadata in RadImageNet, in-

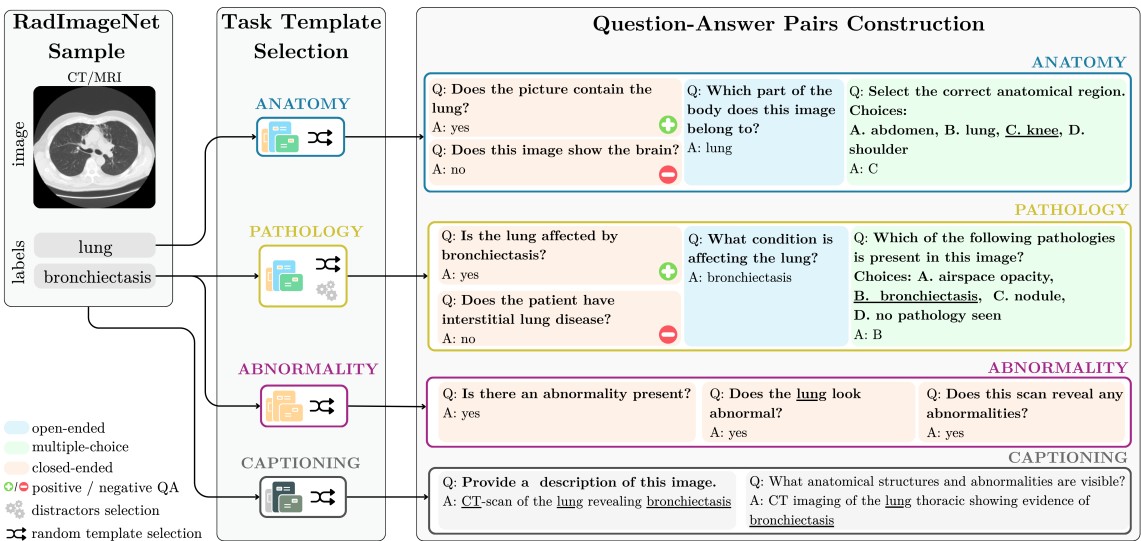

Figure 2: **RadImageNet-VQA construction pipeline.** Expert-annotated CT/MRI images are converted into radiology-aware captions and VQA samples using task- and format-specific templates. The pipeline generates open-ended, closed-ended, and multiple-choice questions across anatomy recognition, abnormality detection, and pathology identification, with distractors designed to prevent shortcuts.

cluding acquisition modality, anatomical region, and pathology category, and convert it into structured radiologic captions. Each image is paired with a synthesized textual description generated from these metadata fields sampling randomly from a diverse set of radiology-aware templates. For example, modality and anatomy tags are verbalized in formulations like `"A [modality] scan of the [anatomy] showing [pathology]"`.

**VQA sample generation.** To support both instruction tuning and benchmark construction, each annotated image is converted into structured VQA samples using task-specific question-answer templates. As illustrated in Figure 2, we generate closed-ended, open-ended, and multiple-choice questions across three complementary tasks: *anatomy recognition*, *abnormality detection* and *pathology identification*. Anatomy recognition identifies the imaged region, abnormality detection determines whether an image contains any abnormal finding, and pathology identification—by far the most challenging—requires distinguishing between specific diseases within a given anatomical context.

For each task, we design a set of template-based question formulations for each question type (Appendix A.3). For every task and question type, we define between 2 and 7 linguistic variations, ensuring diversity while reducing opportunities for models to exploit textual shortcuts. For closed-ended anatomy and pathology questions, we generate both positive and negative variants, where the expected answer is respectively "yes" (the organ or pathology is present) or "no" (it is absent). These paired formulations allow us to probe VLMs more precisely, particularly with respect to biases such as hallucinating organs or pathologies even when they are not present.

Multiple-choice questions require careful construction as distractors, *i.e.* incorrect answer options, must be chosen in a way that prevents trivial elimination. For anatomy

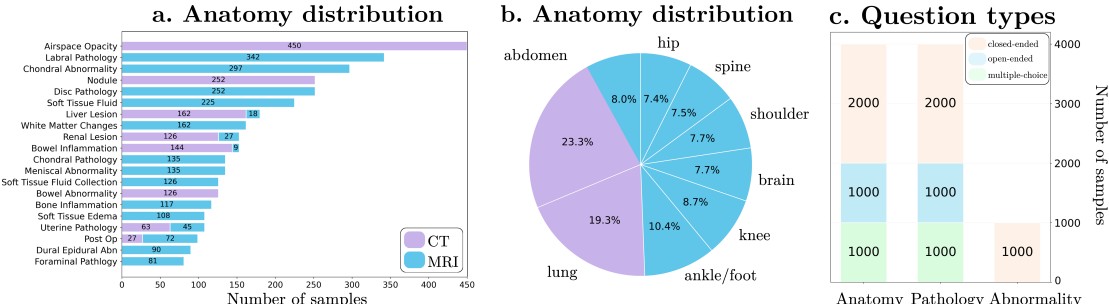

Figure 3: **Composition of the RadImageNet-VQA benchmark**, containing 1,000 CT/MRI images and 9,000 QA pairs: (a) most frequent pathology labels, (b) anatomy distribution, (c) question-type distribution.

recognition, distractors are sampled from other existing anatomical regions in the dataset. For pathology identification, distractors are restricted to clinically plausible diseases from the same region, ensuring that the model cannot solve the question by matching anatomy alone. We also include the option *"no pathology seen"* in every pathology multiple-choice. As shown in Appendix C.1, increasing the frequency of this option in multiple-choice options causes a clear and consistent drop in accuracy for medical-oriented models (e.g., MedGemma and Lingshu), while general-purpose models remain comparatively stable. This suggests that medical VLMs often assume that pathology is present and treat "no pathology" as unlikely instead of confirming it visually. Including this option therefore reduces this bias and encourages models to distinguish true pathology from its absence, resulting in a more challenging and faithful evaluation.

**Training set creation.** We apply the captioning and VQA generation pipeline to the full RadImageNet training split to construct a large-scale corpus for multimodal model training. This yields structured radiologic captions for medical visual-text alignment and extensive VQA data for instruction tuning. In total, the training corpus comprises approximately 750K images with 7.5M samples (750K image-captions and 6.75M QA pairs). A complete list of anatomical regions and pathology categories is provided in Appendix A.1.

### 3.2. Benchmark

For evaluation, we sample 1,000 CT/MRI images from the RadImageNet test split while preserving the distribution of anatomical regions, pathology categories, and abnormal cases. Applying our pipeline yields 9,000 closed-ended, open-ended, and multiple-choice QA pairs. Figure 3 shows the distribution of pathology labels, question formats, anatomical regions, and normal/abnormal cases. Samples are evenly distributed across tasks and question types, and the dataset includes both normal (28.1%) and abnormal (71.9%) studies across multiple anatomical regions. The most common pathology categories include a wide range of soft-tissue, musculoskeletal, and neuro-abdominal findings represented in CT and MRI.

**Evaluation metrics.** Model predictions are scored as correct or incorrect, and mean accuracy is reported. Closed-ended (yes/no) questions are judged by checking for the ex-

| Model | Anatomy | | | | Abn | Pathology | | | | Avg |
|---|---|---|---|---|---|---|---|---|---|---|
| | Open | Closed+ | Closed– | MC | Closed | Open | Closed+ | Closed– | MC | |
| **General-purpose models** | | | | | | | | | | |
| LLaVA-OneVision-Qwen2-7B | 48.4 | 82.7 | 81.3 | 88.7 | 49.8 | 16.0 | 55.3 | 61.3 | 33.6 | 57.5 |
| Qwen2.5-VL-3B-Instruct | 37.7 | 83.7 | 77.1 | 77.9 | 70.5 | 10.0 | 78.1 | 21.4 | 34.8 | 54.6 |
| Qwen2.5-VL-7B-Instruct | 37.5 | 84.9 | 79.1 | 80.5 | 69.5 | 9.8 | 69.2 | 47.4 | 30.1 | 56.4 |
| InternVL3.5-8B | 50.9 | 98.1 | 75.9 | **93.3** | 58.9 | 9.9 | 85.9 | 27.8 | 41.8 | 60.3 |
| InternVL3.5-14B | 56.6 | **98.2** | 74.4 | 89.9 | **74.4** | 11.7 | **86.7** | 33.7 | **47.1** | **63.6** |
| GPT-5 | 44.3 | 72.4 | 81.8 | 89.3 | 27.5 | 15.8 | 54.9 | 68.3 | 41.2 | 54.9 |
| Gemini 2.5 Pro | **65.7** | 76.5 | 81.9 | 88.8 | 17.8 | 21.1 | 50.2 | 30.1 | 44.4 | 52.9 |
| **Medical-oriented models** | | | | | | | | | | |
| LLaVA-Med-v1.5-mistral-7b | 44.3 | 89.9 | 55.3 | 58.1 | 22.4 | 10.2 | 41.8 | 66.6 | 26.4 | 48.2 |
| HuatuoGPT-Vision-7B | 45.4 | 82.5 | 89.0 | 88.3 | 60.6 | 13.6 | 65.5 | 69.2 | 44.6 | 48.9 |
| medgemma-4b-it | 62.9 | 76.4 | 82.5 | 84.8 | 55.4 | **30.6** | 54.2 | 77.4 | 36.8 | 51.5 |
| Lingshu-7B | 49.6 | 90.7 | 85.1 | 88.9 | 47.9 | 15.7 | 57.0 | 78.8 | 29.6 | 60.4 |
| Lingshu-32B | 45.2 | 75.5 | **92.1** | 89.3 | 54.5 | 14.4 | 46.4 | **88.8** | 31.7 | 59.8 |

Table 2: **Zero-shot accuracies (%) of VLMs on RadImageNet-VQA benchmark.** Results are reported across anatomy recognition, abnormality detection (*Abn*), and pathology identification using four question formats: *Open* (free-form), *Closed+* (always 'yes' as true answer) , *Closed–* (always 'no'), and *MC* (multiple-choice).

pected token. Open-ended responses are evaluated using *LLM-as-a-judge* (Zheng et al., 2023) framework against ground-truth (Appendix B.1). For multiple-choice questions, we extract the predicted option letter with rule-based parser and compare it to the true answer.

To assess the reliability of the LLM judge on open-ended samples, we conducted a human validation on a stratified subset of 380 examples, balanced across the 10 evaluated models and LLM judge outcomes. Each example (question, ground-truth answer, and model response) correctness was independently annotated by two reviewers, blinded to both model identity and LLM judge decisions. The LLM judge matches human evaluation in 90.6% of cases, achieving a substantial agreement with a Cohen's $\kappa$ score of 0.72, supporting its use for open-ended evaluation in the benchmark. Validation details are provided in Appendix B.2.

## 4. Experiments

We evaluate state-of-the-art VLMs on RadImageNet-VQA under zero-shot and fine-tuned settings, and analyze text shortcuts, data-sampling strategies, and vision encoder choices.

### 4.1. Zero-shot evaluation

We benchmark a wide range of general-purpose and medical-oriented VLMs, including open-source models (e.g., LLaVA-OneVision, Qwen2.5-VL, InternVL) and proprietary systems (e.g., GPT-5, Gemini 2.5 Pro) on the full RadImageNet-VQA benchmark. Models generate free-text answers for open-ended questions and letter choices for multiple-choice questions. Accuracy is computed via exact match for closed and MC questions and LLM-as-a-judge scoring for open-ended responses. Results are displayed in Table 2.

First, pathology identification emerges as the primary bottleneck. While models perform well on anatomy recognition (e.g., InternVL3.5-8B (Chen et al., 2024) achieves 93.3% on anatomy multiple-choice), they struggle severely with fine-grained disease identification. Open-ended pathology questions prove particularly challenging, with most models scoring below 20% accuracy, and even the best performer on this task, MedGemma-4b (Sellergren et al., 2025), attains only 30.6%. These results underscore that discriminating between specific pathologies based on visual features remains a major unsolved problem.

Then, we observe a nuanced performance gap between general-purpose and medical-oriented models. General-purpose models achieve the highest average accuracy with InternVL3.5 -14B (63.6%) and lead in several categories, including abnormality detection and closed-ended pathology questions. Medical-oriented models outperform general ones only in some specific areas, such as MedGemma's superior performance on open-ended pathology questions. This suggests that scale and broad visual–language pretraining still provide a stronger foundation than existing medical specialization strategies.

Finally, the results reveal that even the most capable general-purpose models lack essential radiologic skills. GPT-5 (27.5%) and Gemini 2.5 Pro (17.8%) perform poorly on abnormality detection, below random guessing. We intuit that aggressive safety alignment may suppress the model's willingness to identify abnormalities, causing them to default to conservative "no abnormality" responses rather than rely on visual evidence.

### 4.2. Text shortcuts analysis

To assess whether RadImageNet-VQA can be solved from linguistic priors alone, we evaluate several VLMs in a text-only setting, where images are removed and models receive only the question. We compare performance on VQA-RAD (Lau et al., 2018), SLAKE (Liu et al., 2021), MMMU-Med-val (Yue et al., 2024), and RadImageNet-VQA benchmarks to quantify the extent to which different datasets allow text-based shortcutting. (see Figure 4)

On VQA-RAD and SLAKE, models reach 11.5–33% text-only accuracy, indicating that many answers can be partially inferred from linguistic regularities. By contrast, open-ended accuracy on RadImageNet-VQA drops to near-random levels (2–10%) across all models, confirming that shortcut cues are largely removed. The text-only evaluation also reveals differences in model behavior. Some models, e.g. Lingshu-7B and InternVL variants, maintain non-zero accuracy on RadImageNet-VQA, but qualitative inspection shows that these

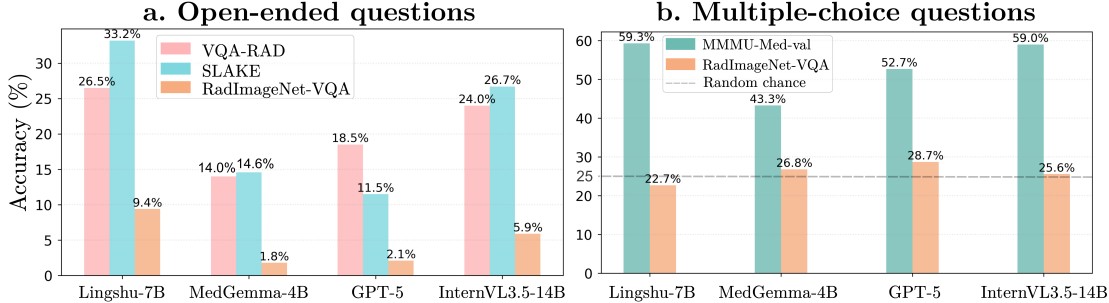

Figure 4: **Text-only analysis of multiple VLMs' accuracy** for open-ended and MC questions on RadImageNet-VQA, VQA-RAD, SLAKE, and MMU-Med-val.

predictions reflect dataset priors rather than uncertainty awareness. Models rarely acknowledge missing visual information and instead commit to plausible guesses based on high-frequency anatomical regions or semantic variants of pathology terms, rather than image-grounded reasoning. Some examples are presented in Appendix C.6.

The same pattern holds for multiple-choice questions. On MMMU-Med-val, text-only accuracy remains noticeably above random, reflecting reliance on textual priors or weak distractor design. In contrast, RadImageNet-VQA collapses to the expected 25% baseline, indicating that models cannot exploit option distributions or question phrasing. We can further observe in Appendix 13 that residual correct answers are largely driven by conservative defaults—such as selecting "no pathology seen" when no image is available—which yield correct predictions on normal cases but fail on abnormal ones. General-purpose models show a flatter spread across options, consistent with near-random guessing.

Overall, these findings show that RadImageNet-VQA substantially suppresses linguistic shortcuts in both open-ended and multiple-choice formats. When deprived of images, models fail to recover anatomical or pathological information and instead rely on dataset frequency or generic medical priors. This confirms that RadImageNet-VQA requires image-grounded interpretation rather than text-driven guessing.

### 4.3. Fine-tuning on RadImageNet-VQA

#### 4.3.1. EXPERIMENTAL SETUP

As training data, we assemble a multimodal corpus combining: (1) RadImageNet-VQA train split, (2) CT/MRI datasets converted into 2D VQA pairs– KiTS22 (Cardoso et al., 2017) and AbdomenAtlas (Chen et al., 2025a))– and (3) existing radiologic VQA datasets –VQA-RAD (Lau et al., 2018), SLAKE (Liu et al., 2021)–, and the radiology subset of LLaVA-Med (Li et al., 2023a). Additional details and workflow are provided in Appendix D.

We fine-tune two widely used open-source VLMs, LLaVA-OneVision (Li et al., 2024) and Qwen2.5-VL-7B-Instruct (Bai et al., 2025), and a medical-oriented model, Lingshu-7B (Xu et al., 2025b). All models employ of a SigLIP (Zhai et al., 2023) vision encoder that produces image embeddings. LLaVA-OneVision pairs it with a Qwen2-7B (Yang et al., 2024) language model while Qwen2.5-VL-7B is based on the more re-

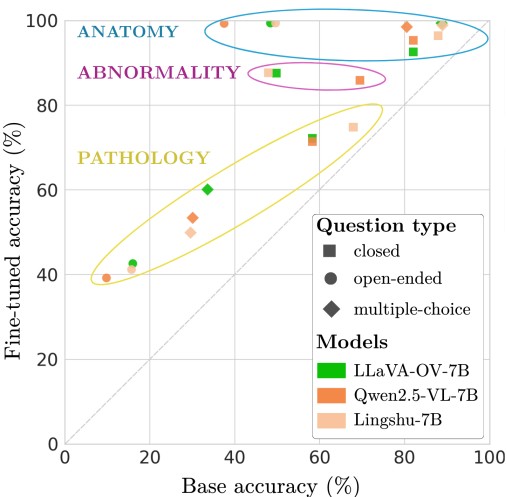

Figure 5: Comparison of base and fine-tuned accuracies on RadImageNet-VQA.

cent Qwen2.5-7B language model. Lingshu-7B share the same architecture as Qwen2.5-VL-7B but has been pretrained and fine-tuned on data including medical data. This allows us to apply the same training recipes across architectures while comparing general-purpose versus medically-aligned initialization.

We follow the standard two-phase training paradigm used in recent multimodal tuning consisting of (i) an *alignment stage*, where the LLM is frozen and only the vision encoder and projection layers are updated to adapt to radiologic features; and (ii) *instruction tuning*, where the full model is fine-tuned on supervised VQA data.

### 4.3.2. Evaluation of fine-tuned models

Fine-tuning on the composed radiologic corpus yields consistent and substantial performance gains across all models. As shown in Figure 5, every point shifts above the diagonal, indicating reliable gains regardless of architecture, with average increases of +19.5% to +22.5% (Appendix 11). Task-level results reveal that anatomy recognition is nearly saturated after fine-tuning (98.5–99.4% on multiple-choice), abnormality detection shows the largest relative gains (+16.4–39.8%, reaching 85.9 – 87.7%), while pathology identification remains the primary bottleneck, clustering at lower accuracy even post-training. These trends hold across model families, demonstrating both the generality of our training strategy and the limits of class-level supervision for fine-grained disease identification. Finally, Lingshu-7B and Qwen2.5-VL-7B converge to nearly identical performance, indicating that radiologic supervision, and not prior medical pretraining, is the main driver of downstream capability.

### 4.3.3. Ablations on vision encoder and data sampling

We compare four variants of LLaVA-OneVision variants using the standard SigLIP encoder or MedSigLIP (Sellergren et al., 2025), and with two different data-sampling strategies during fine-tuning. *Mixed sampling* blends samples from all datasets within each batch, whereas *alternating sampling* draws each batch from a single dataset and the source alternates across batches to balance exposure over time. Performance is measured across VQA-RAD, SLAKE, and RadImageNet-VQA and reported in Figure 6. Models initialized with standard SigLIP outperform those using the medically pre-trained MedSigLIP, which may indicate that pretraining with multiple sources of medical images, including textbooks and X-ray, does not translate into systematic gains for VQA with CT and MRI. Alternating sampling data strategy yields slightly stronger and more stable results than mixed sampling, particularly early in training, likely because it reduces inter-dataset interference under small-batch constraints.

## 4.4. Qualitative Analysis

To better understand model behavior on open-ended questions, Figure 7 presents some qualitative examples comparing anatomy recognition and pathology identification. Anatomy recognition is generally robust across models, especially for large, well-defined structures (e.g., shoulder, lung, ankle/foot) where shape and contextual cues are strong. For broader regions such as the abdomen, zero-shot models sometimes predict a salient sub-organ (e.g., bladder/uterus) rather than the full region.

In contrast, pathology identification remains the primary bottleneck. Models frequently misidentify or overlook subtle, localized, or rare abnormalities. For example, small or fine-grained lesions such as *patella pathology*, *coalition bone fusion*, or *quadriceps pathology* are missed or confuse with surrounding structures by many zero-shot models (e.g., GPT-5,

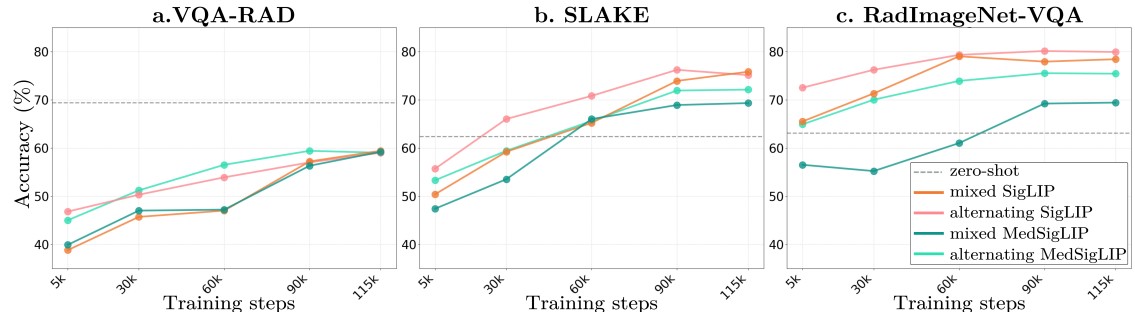

Figure 6: **Impact of vision encoder and sampling with LLaVA-OneVision.** Alternating sampling with standard SigLIP yields the strongest performance.

MedGemma-4B, Lingshu-32B). Similarly, *osseous abnormalities* in the spine or *infraspinatus/rotator cuff injuries* were often misclassified or described ambiguously, highlighting texture ambiguity and small-object difficulty. Even visually salient findings such as *liver lesions* or *airspace opacities* can be inconsistently labeled across models, suggesting sensitivity to appearance variation. Fine-tuning improves recognition for some categories (e.g., *chondral abnormalities*, *liver lesions*), but rare or subtle pathologies like *Lisfranc ligament injury*, and *ACL tears*, or *coalition bone fusion* remain challenging.

## 5. Discussion

RadImageNet-VQA provides large-scale CT and MRI supervision for radiologic VQA, spanning abnormality detection, anatomy recognition, and fine-grained pathology identification across multiple question formats. Benchmark results show that current VLMs perform relatively well on anatomy and abnormality recognition, while fine-grained pathology identification remains the main failure mode even after fine-tuning. Together with the text-only ablations, these results suggest that progress will require better visual grounding for subtle findings rather than exploiting linguistic priors. Beyond benchmarking, the dataset serves as a diagnostic tool for model behavior and a scalable fine-tuning resource for VLMs. Additionally, the inclusion of both CT and MRI also enables future cross-modality studies.

**Limitations.** The dataset inherits several constraints from the underlying RadImageNet taxonomy, most notably a single-label pathology per image (Mei et al., 2022). This setting does not fully reflect the complexity of some real-world clinical cases which can involve multiple co-existing findings. Hence, promising extensions include adding samples with multi-finding annotations and richer clinical context. Potential biases such as pathology prevalence and imaging protocols may also disadvantage underrepresented conditions. Beyond these inherited aspects, our formulation has additional limitations: (1) questions are programmatically generated, which increases control and scalability but may underrepresent real clinical language and reasoning; (2) the dataset is 2D, so models do not have access to volumetric context; (3) open-ended evaluation relies on an automatic judge, which we validate but may still introduce residual uncertainty for borderline cases.

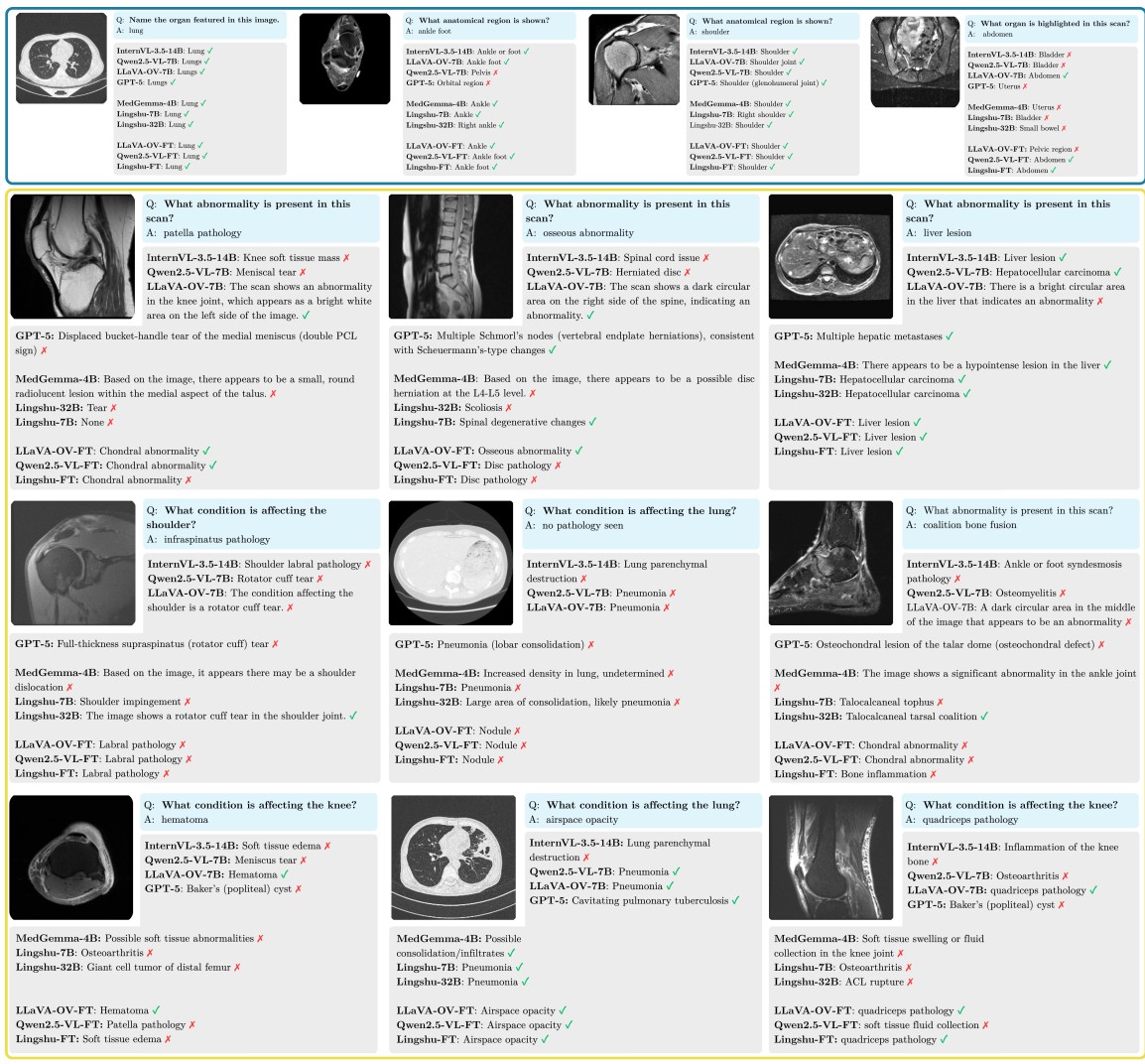

Figure 7: Qualitative examples of model predictions for anatomy recognition (*top*) and pathology identification (*bottom*). Each row shows one case with ground-truth; predictions are marked as correct (✓) or incorrect (✗) according to LLM-Judge.

## 6. Conclusion

In this work, we introduce RadImageNet-VQA, a large-scale CT/MRI dataset for radiologic VQA built through a rigorous QA-generation pipeline. It includes both a training set and a curated benchmark, covering multiple tasks and anatomy regions. Our experiments show that fine-grained pathology identification remains challenging for current models. Fine-tuning current VLMs on RadImageNet-VQA yields substantial performance gains, while greatly reducing text-based shortcuts, which demonstrate its potential to be a valuable resource for developing and assessing stronger medical VLMs for radiology.

## Acknowledgment

We thank the RadImageNet team for granting access to the RadImageNet dataset and for making this resource available for research purposes (https://www.radimagenet.com/). We thank the Jean Zay and Leonardo HPC centers for providing the GPU computing resources used in this work.

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

# Appendix A. Dataset details

## A.1. RadImageNet-VQA anatomy regions and pathology taxonomy

The full taxonomy of CT and MRI anatomical regions paired with fine-grained pathology labels used for caption and VQA generation is reported in Table 3.

| Anatomy | Pathologies |
|---------|-------------|
| abdomen | abnormal entire organ, adrenal pathology, arterial pathology, ascites, biliary dilation, bladder pathology, bowel abnormality, bowel inflammation, bowel mass, degenerative changes, dilated urinary tract, enlarged organ, fat-containing tumor, gallbladder pathology, gallstone, intraperitoneal mass, liver disease, liver lesion, marrow abnormality, osseous neoplasm, ovarian pathology, pancreatic lesion, post-operative state, prostate lesion, renal lesion, soft tissue collection, soft tissue mass, splenic lesion, urolithiasis, uterine pathology |
| ankle foot | achilles pathology, anterior talofibular ligament pathology, bone inflammation, calcaneofibular ligament pathology, chondral abnormality, coalition (bone fusion), deltoid pathology, extensor pathology, fat-containing tumor, flexor pathology, hematoma, intra-articular abnormality, Lisfranc joint injury, osseous disruption, osseous neoplasm, peroneal tendon pathology, plantar fascia pathology, plantar plate tear, post-operative state, soft tissue edema, soft tissue fluid, soft tissue mass, spring ligament injury, syndesmosis pathology |
| brain | acute infarct, arteriovenous anomaly, chronic infarct, edema, extra-axial lesion, focal FLAIR hyperintensity, intra-articular abnormality, pituitary lesion, white matter changes |
| hip | abductor pathology, capsular pathology, chondral pathology, hamstring pathology, hematoma, labral pathology, marrow inflammation, osseous disruption, osseous lesion, post-operative state, soft tissue edema, soft tissue fluid, soft tissue mass |
| knee | anterior cruciate ligament pathology, bone inflammation, chondral abnormality, fibular collateral ligament pathology, fracture, hematoma, intra-articular abnormality, medial collateral ligament pathology, meniscal abnormality, muscle strain, patella pathology, post-operative ACL reconstruction, posterior cruciate ligament pathology, quadriceps pathology, soft tissue edema, soft tissue fluid collection, soft tissue mass |
| lung | airspace opacity, bronchiectasis, interstitial lung disease, nodule, parenchyma destruction |
| shoulder | acromioclavicular joint osteoarthritis, biceps pathology, calcific tendinosis, glenohumeral joint osteoarthritis, infraspinatus pathology, labral pathology, marrow inflammation, osseous lesion, post-operative state, soft tissue edema, soft tissue fluid, subscapularis pathology, supraspinatus pathology |
| spine | cystic lesions, disc pathology, dural or epidural abnormality, facet arthropathy, foraminal pathology, osseous abnormality, scoliosis, spinal cord pathology |

Table 3: Anatomical regions and corresponding pathologies in RadImageNet-VQA

## A.2. RadImageNet-VQA anatomy regions and pathology distribution

Table 4: **Anatomy region distribution** in train corpus and benchmark, sorted by decreasing training-set frequency. Percentages are computed over images. Δpp are percentages differences in points of benchmark compared to train.

| Anatomy | Train count (%) | Benchmark count (%) | Δpp |
|---|---|---|---|
| Abdomen | 189,104 (25.21%) | 313 (31.30%) | +6.09 |
| Ankle Foot | 146,009 (19.47%) | 104 (10.40%) | -9.07 |
| Knee | 145,493 (19.40%) | 87 (8.70%) | -10.70 |
| Lung | 89,709 (11.96%) | 193 (19.30%) | +7.34 |
| Spine | 57,038 (7.60%) | 75 (7.50%) | -0.10 |
| Shoulder | 42,669 (5.69%) | 77 (7.70%) | +2.01 |
| Hip | 42,480 (5.66%) | 74 (7.40%) | +1.74 |
| Brain | 37,507 (5.00%) | 77 (7.70%) | +2.70 |

Table 5: **Pathology distribution.** in train corpus and benchmark, , sorted by decreasing training-set frequency. Percentages are computed over images. Δpp are percentages differences in points of benchmark compared to train..

| Pathology | Train count (%) | Benchmark count (%) | Δpp |
|---|---|---|---|
| Normal | 196,507 (26.20%) | 281 (28.10%) | +1.90 |
| Chondral Abnormality | 76,222 (10.16%) | 33 (3.30%) | -6.86 |
| Meniscal Abnormality | 36,414 (4.86%) | 15 (1.50%) | -3.36 |
| Soft Tissue Fluid | 33,461 (4.46%) | 25 (2.50%) | -1.96 |
| Labral Pathology | 32,744 (4.37%) | 38 (3.80%) | -0.57 |
| Bone Inflammation | 32,508 (4.33%) | 13 (1.30%) | -3.03 |
| Airspace Opacity | 30,861 (4.11%) | 50 (5.00%) | +0.89 |
| Disc Pathology | 26,415 (3.52%) | 28 (2.80%) | -0.72 |
| Nodule | 21,047 (2.81%) | 28 (2.80%) | -0.01 |
| Soft Tissue Fluid Collection | 20,781 (2.77%) | 14 (1.40%) | -1.37 |
| Soft Tissue Edema | 14,178 (1.89%) | 12 (1.20%) | -0.69 |
| Liver Lesion | 11,714 (1.56%) | 20 (2.00%) | +0.44 |
| Renal Lesion | 11,275 (1.50%) | 17 (1.70%) | +0.20 |
| Osseous Disruption | 10,655 (1.42%) | 7 (0.70%) | -0.72 |
| ATFL Pathology | 9,698 (1.29%) | 6 (0.60%) | -0.69 |
| Foraminal Pathology | 9,674 (1.29%) | 9 (0.90%) | -0.39 |
| Post-operative Changes | 9,009 (1.20%) | 11 (1.10%) | -0.10 |
| ACL Pathology | 8,191 (1.09%) | 4 (0.40%) | -0.69 |
| White Matter Changes | 8,166 (1.09%) | 18 (1.80%) | +0.71 |

| Pathology | Train count (%) | Benchmark count (%) | Δpp |
|---|---|---|---|
| Bowel Abnormality | 7,966 (1.06%) | 14 (1.40%) | +0.34 |
| Interstitial Lung Disease | 7,872 (1.05%) | 8 (0.80%) | -0.25 |
| Soft Tissue Mass | 7,763 (1.04%) | 6 (0.60%) | -0.44 |
| Achilles Pathology | 7,413 (0.99%) | 4 (0.40%) | -0.59 |
| Dural/Epidural Abnormality | 6,668 (0.89%) | 10 (1.00%) | +0.11 |
| Chondral Pathology | 6,598 (0.88%) | 15 (1.50%) | +0.62 |
| MCL Pathology | 6,427 (0.86%) | 4 (0.40%) | -0.46 |
| Peroneal Pathology | 6,017 (0.80%) | 4 (0.40%) | -0.40 |
| Bowel Inflammation | 5,910 (0.79%) | 17 (1.70%) | +0.91 |
| Intracranial Pathology | 5,338 (0.71%) | 5 (0.50%) | -0.21 |
| Uterine Pathology | 5,196 (0.69%) | 12 (1.20%) | +0.51 |
| Plantar Fascia Pathology | 5,183 (0.69%) | 6 (0.60%) | -0.09 |
| Supraspinatus Pathology | 4,197 (0.56%) | 5 (0.50%) | -0.06 |
| Marrow Inflammation | 3,853 (0.51%) | 7 (0.70%) | +0.19 |
| CFL Pathology | 3,395 (0.45%) | 4 (0.40%) | -0.05 |
| Osseous Neoplasm | 3,324 (0.44%) | 6 (0.60%) | +0.16 |
| Flexor Pathology | 2,915 (0.39%) | 4 (0.40%) | +0.01 |
| Pancreatic Lesion | 2,646 (0.35%) | 7 (0.70%) | +0.35 |
| Ovarian Pathology | 2,633 (0.35%) | 5 (0.50%) | +0.15 |
| Glenohumeral Joint OA | 2,378 (0.32%) | 5 (0.50%) | +0.18 |
| Deltoid Pathology | 2,209 (0.29%) | 4 (0.40%) | +0.11 |
| Osseous Lesion | 2,174 (0.29%) | 6 (0.60%) | +0.31 |
| Acromioclavicular Joint OA | 2,158 (0.29%) | 3 (0.30%) | +0.01 |
| Adrenal Pathology | 2,101 (0.28%) | 5 (0.50%) | +0.22 |
| Gallstone | 1,923 (0.26%) | 7 (0.70%) | +0.44 |
| Osseous Abnormality | 1,911 (0.25%) | 3 (0.30%) | +0.05 |
| Ascites | 1,853 (0.25%) | 8 (0.80%) | +0.55 |
| Chronic Infarct | 1,842 (0.25%) | 7 (0.70%) | +0.45 |
| Scoliosis | 1,655 (0.22%) | 4 (0.40%) | +0.18 |
| Bladder Pathology | 1,599 (0.21%) | 5 (0.50%) | +0.29 |
| Parenchymal Destruction | 1,519 (0.20%) | 5 (0.50%) | +0.30 |
| Bronchiectasis | 1,408 (0.19%) | 4 (0.40%) | +0.21 |
| Urolithiasis | 1,378 (0.18%) | 8 (0.80%) | +0.62 |
| Biceps Pathology | 1,323 (0.18%) | 5 (0.50%) | +0.32 |
| Intraperitoneal Mass | 1,303 (0.17%) | 3 (0.30%) | +0.13 |
| Fracture | 1,275 (0.17%) | 3 (0.30%) | +0.13 |
| Arterial Pathology | 1,154 (0.15%) | 6 (0.60%) | +0.45 |
| Patella Pathology | 1,147 (0.15%) | 3 (0.30%) | +0.15 |
| Cystic Lesions | 1,104 (0.15%) | 3 (0.30%) | +0.15 |
| Extrastructural Pathology | 1,020 (0.14%) | 7 (0.70%) | +0.56 |
| Dilated Urinary Tract | 955 (0.13%) | 8 (0.80%) | +0.67 |
| Liver Disease | 868 (0.12%) | 3 (0.30%) | +0.18 |
| Prostate Lesion | 835 (0.11%) | 3 (0.30%) | +0.19 |
| Biliary Dilation | 833 (0.11%) | 3 (0.30%) | +0.19 |

| Pathology | Train count (%) | Benchmark count (%) | Δpp |
|---|---|---|---|
| Abductor Pathology | 724 (0.10%) | 3 (0.30%) | +0.20 |
| Splenic Lesion | 718 (0.10%) | 3 (0.30%) | +0.20 |
| Hematoma | 642 (0.09%) | 4 (0.40%) | +0.31 |
| Marrow Abnormality | 600 (0.08%) | 3 (0.30%) | +0.22 |
| Focal FLAIR Hyperintensity | 592 (0.08%) | 6 (0.60%) | +0.52 |
| PCL Pathology | 585 (0.08%) | 4 (0.40%) | +0.32 |
| Spinal Cord Pathology | 580 (0.08%) | 3 (0.30%) | +0.22 |
| Fat-containing Tumor | 540 (0.07%) | 3 (0.30%) | +0.23 |
| Plantar Plate Tear | 536 (0.07%) | 3 (0.30%) | +0.23 |
| Extensor Pathology | 536 (0.07%) | 3 (0.30%) | +0.23 |
| Gallbladder Pathology | 517 (0.07%) | 3 (0.30%) | +0.23 |
| Bowel Mass | 460 (0.06%) | 3 (0.30%) | +0.24 |
| Muscle Strain | 424 (0.06%) | 3 (0.30%) | +0.24 |
| Acute Infarct | 423 (0.06%) | 3 (0.30%) | +0.24 |
| Quadriceps Pathology | 405 (0.05%) | 3 (0.30%) | +0.25 |
| FCL Pathology | 367 (0.05%) | 3 (0.30%) | +0.25 |
| Syndesmosis Pathology | 318 (0.04%) | 3 (0.30%) | +0.26 |
| Facet Arthropathy | 302 (0.04%) | 3 (0.30%) | +0.26 |
| Soft Tissue Collection | 238 (0.03%) | 3 (0.30%) | +0.27 |
| Arteriovenous Anomaly | 217 (0.03%) | 3 (0.30%) | +0.27 |
| Global Organ Abnormality | 211 (0.03%) | 3 (0.30%) | +0.27 |
| Degenerative Changes | 197 (0.03%) | 3 (0.30%) | +0.27 |
| Hamstring Pathology | 192 (0.03%) | 3 (0.30%) | +0.27 |
| Organomegaly | 170 (0.02%) | 3 (0.30%) | +0.28 |
| Subscapularis Pathology | 115 (0.02%) | 3 (0.30%) | +0.28 |
| Capsular Pathology | 99 (0.01%) | 3 (0.30%) | +0.29 |
| Edema | 98 (0.01%) | 3 (0.30%) | +0.29 |
| Spring Ligament Injury | 83 (0.01%) | 3 (0.30%) | +0.29 |
| Infraspinatus Pathology | 76 (0.01%) | 3 (0.30%) | +0.29 |
| Post-operative ACL | 71 (0.01%) | 3 (0.30%) | +0.29 |
| Pituitary Lesion | 64 (0.01%) | 3 (0.30%) | +0.29 |
| Calcific Tendinosis | 63 (0.01%) | 3 (0.30%) | +0.29 |
| Tarsal Coalition | 50 (0.01%) | 3 (0.30%) | +0.29 |
| Lisfranc Pathology | 37 (0.00%) | 3 (0.30%) | +0.30 |

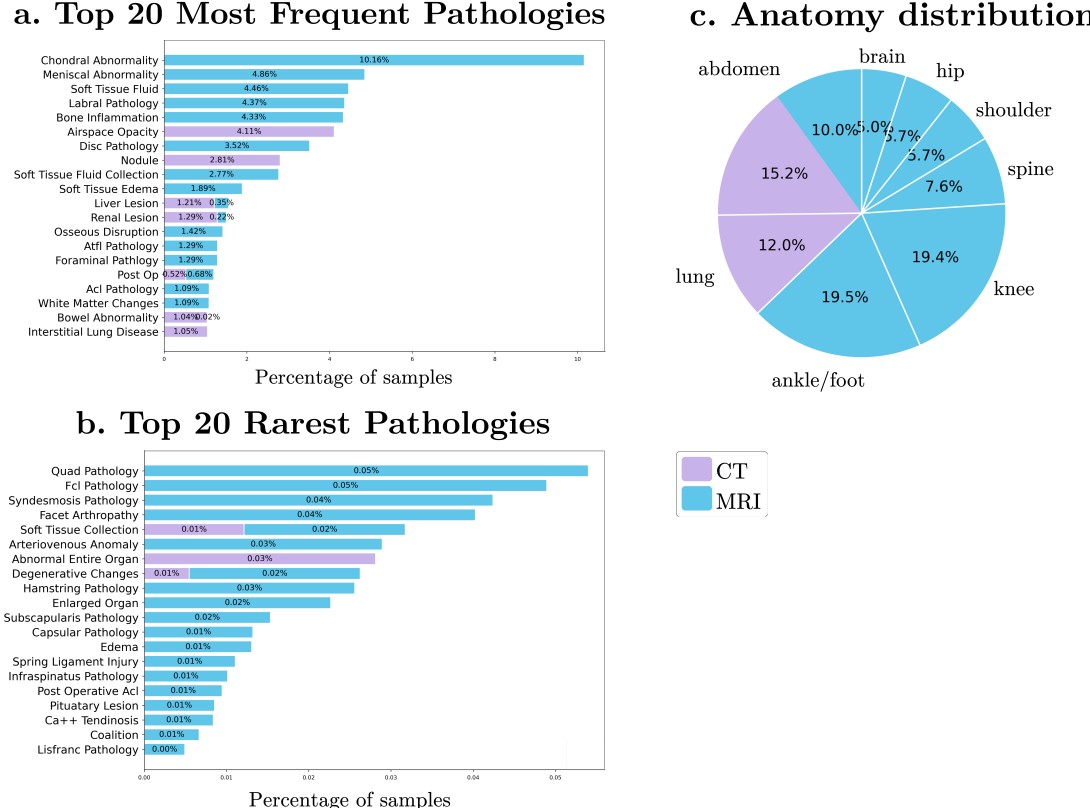

Figure 8: **Composition of the RadImageNet-VQA training set**, containing 750k CT/MRI images : (a) most frequent pathology labels, (b) less frequent pathology labels, (c) anatomy distribution. Percentages are computed over images.

## A.3. Template set for VQA and captioning generation

Figure 9 illustrates the template families used to generate radiologic captions and VQA samples for all tasks and question types.

### ANATOMY

- Does the picture contain the {anatomy}?
- Does this image show the {anatomy}?
- Is the {anatomy} part of this scan?
- Is the {anatomy} visible in the image?
- Can you see the {anatomy}?
- Is the {anatomy} present in this image?
- Can you identify the {anatomy} in this image?

- Which body part is visible in this image?
- Which part of the body does this image belong to?
- What organ is highlighted in this scan?
- What anatomical region is shown?
- What specific anatomical structure is highlighted?
- Name the organ featured in this image.

- Which of the following anatomical regions is depicted in this scan?
- What is the correct anatomical region for this scan?
- Select the correct anatomical region.

### ABNORMALITY

- Is this image abnormal?
- Is there an abnormality present?
- Does this scan reveal any abnormalities?
- Does the {anatomy} look abnormal?

### PATHOLOGY

- Is there evidence of {pathology} in this image?
- Does the patient have {pathology}?
- Is {pathology} present?
- Are there signs of {pathology}?
- Is the {anatomy} affected by {pathology}?

- What abnormality is present in this scan?
- What condition is affecting the {anatomy}?

- Which of the following pathologies is present in this image?
- What is the most likely pathology in this image?
- Select the pathology that best describes this image:
- What pathology is shown in this scan of {anatomy}

### CAPTIONING

- Provide a thorough clinical description of this medical image.
- What findings and structures are visible in this scan?
- Describe the medical imaging features and anatomical details.
- Give a detailed diagnostic description of this image.
- What does this scan show in terms of anatomy and pathology?
- Describe the radiological findings and anatomical structures.
- Provide a comprehensive clinical description of this scan.
- What imaging characteristics and findings are present?
- Describe the diagnostic features and anatomical structures.
- Give a detailed medical imaging description of this scan.
- What anatomical and pathological findings are demonstrated?
- Describe the clinical imaging findings in this medical scan.
- Provide a complete radiological description of this image.
- What structures, abnormalities, and findings are visible?
- Describe the medical imaging characteristics and clinical findings.
- What does this image reveal about the anatomy and pathology?
- Describe the imaging findings, anatomical structures, and clinical features.
- Provide a detailed clinical and radiological description of this scan.
- What anatomical structures, pathological findings, and imaging features are shown?
- Describe the diagnostic imaging findings and anatomical characteristics.
- Give a comprehensive medical imaging description of this clinical scan.
- What anatomical features, pathological changes, and imaging characteristics are visible?
- Describe the clinical imaging findings, anatomical structures, and diagnostic features.
- Give a detailed medical description of the imaging findings and anatomical details.
- What does this scan reveal about the anatomical structures and pathological findings?
- Describe the diagnostic imaging characteristics, anatomical features, and clinical findings.
- Provide a comprehensive radiological description of the medical imaging findings.
- What anatomical structures, pathological abnormalities, and imaging features are demonstrated?
- Describe the diagnostic imaging characteristics of this scan.

- Describe this medical image.
- Provide a detailed description of this scan.
- What do you see in this medical image?
- Give a comprehensive description of this medical scan.
- What is shown in this medical image?
- Describe the contents of this scan.
- Provide a clinical description of this image.
- What can you observe in this medical image?
- Give a detailed account of what this image shows.
- Describe the imaging findings in this medical image.
- Provide a thorough description of this medical scan.
- What structures and findings are visible in this image?
- Give an overview of what this scan reveals.
- Describe the radiological findings in this image.
- Provide a complete description of this medical image.
- What anatomical and pathological features are shown?
- Describe the imaging characteristics of this scan.
- Give a detailed medical description of this image.
- What can be observed in this diagnostic image?
- Provide a comprehensive overview of this image.
- What does this scan show anatomically and pathologically?
- Describe the medical imaging features present.
- What anatomical structures and abnormalities are visible?
- Describe the radiological characteristics of this image.
- Give a thorough medical description of this scan.
- What anatomical and pathological details are shown?
- Describe the imaging findings and anatomical structures.
- Give a comprehensive medical description of this scan.
- What structures and findings can be identified?
- Describe the anatomical and pathological features.
- Provide a detailed radiological description of this image.
- Describe the clinical and anatomical features shown.
- Give a complete description of the medical imaging findings.
- Describe the clinical findings, anatomical structures, and diagnostic imaging characteristics.
- Give a thorough medical imaging description of the anatomical and pathological features.
- What anatomical structures and abnormalities are demonstrated?

Figure 9: Templates of questions for VQA samples and radiologic captions.

## Appendix B. LLM-as-a-judge evaluation

### B.1. Protocol of evaluation

We adopt the MedEvalKit framework[1] released with Lingshu-7B (Xu et al., 2025b) and use Mistral-Large 2.1 (AI, 2024) as the judge model.

```
Your task is to determine whether the user's answer is correct based on the provided questions
and standard answers (for example, if the user expresses a similar meaning to the standard
answer, or another interpretation of the standard answer, it is considered correct.)
The question is: {question}
The standard answer: {answer} The user's answer: {response}
Please strictly follow the following format for output (0 represents correct, 1 represents
incorrect): <think> your concise think step </think>   <judge>0/1</judge>
For example:
<think> The standard answer is right, and the user's answer is right frontal lobe; they express
the same meaning, so it is correct.  </think> <judge>0</judge>
```

Figure 10: Judgment prompt used in the LLM-as-a-judge module.

### B.2. Validation of LLM-judge

Table 6 presents the validation results where two independent reviewers evaluated 380 stratified samples. Reviewers could mark cases as *correct*, *incorrect*, *don't know*, or *response more precise than ground truth*. For the reported metrics, cases marked as *response more precise than ground truth* were treated as correct, while *don't know* cases (∼6% of total) were conservatively treated as incorrect.

| Reviewer | Agreement (%) | | Cohen's $\kappa$ |
|---|---|---|---|
| | Anatomy | Pathology | |
| Reviewer 1 | 94.20% | 81.05% | 0.7273 |
| Reviewer 2 | 93.98% | 77.27% | 0.7135 |

Table 6: Validation of LLM judge against human reviewers on 380 random samples from various models.

## Appendix C. Additional Zero-Shot Analysis

### C.1. Effect of "no pathology seen" distractors in pathology MC questions

To assess how the construction of multiple-choice distractors influences model behavior on RadImageNet-VQA, we vary the proportion of pathology questions whose answer set includes the option "no pathology seen". We evaluate four representative VLMs—two medically oriented models (Lingshu-7B, MedGemma-4B-it) and two general-purpose models (InternVL3.5-8B, Qwen2.5-VL-7B)—under three configurations in which this option appears in 0%, 30%, or 100% of pathology multiple-choice questions.

---

[1]. https://github.com/alibaba-damo-academy/MedEvalKit

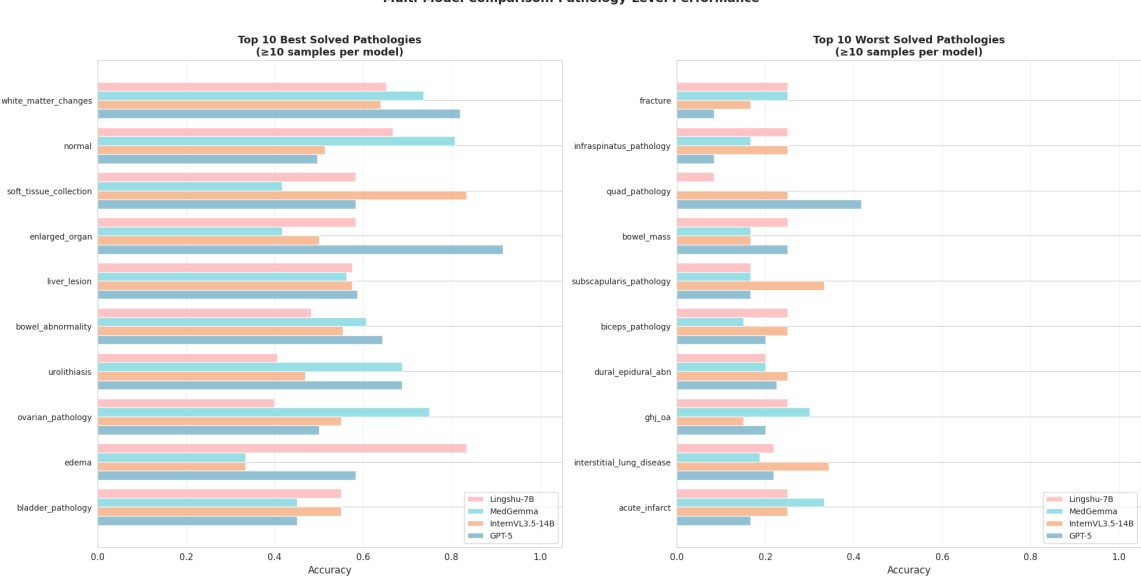

Figure 11: Per-pathology multiple-choice accuracy across four VLMs on RadImageNet-VQA. *Left*: 10 best-solved pathologies; *right*: 10 worst-solved

As reported in Table 7, MedGemma and Lingshu show a progressive decrease in accuracy as the "no pathology" choice becomes more frequent, suggesting that they tend to assume the presence of pathology even when presented with an explicit normal alternative. In contrast, InternVL and Qwen remain comparatively stable or slightly improve, indicating that general models are more inclined to select this choice when uncertain or to rely on deduction from the visible options. Overall, these results show that distractor design can substantially alter model responses and that explicitly modelling the possibility of normal findings remains important for robust radiologic VQA.

| Model | 0% | 30% | 100% |
|---|---|---|---|
| **General-purpose models** | | | |
| InternVL3.5_8B | 45.0 | 44.7 | 46.5 |
| Qwen2.5-VL-7B | 40.5 | 40.3 | 42.6 |
| **Medical specialized models** | | | |
| Lingshu-7B | 59.2 | 54.9 | 49.1 |
| medgemma-4b-it | 60.9 | 57.6 | 50.2 |

Table 7: Accuracy on pathology MC questions when varying the frequency of the "no pathology seen" option.

## C.2. Per-pathology performance analysis

We observe in Table 2 that pathology identification is the most challenging task in the zero-shot setting; to further understand which specific conditions drive this difficulty, Figure 11 reports pathology-level performance by listing the ten best- and worst-solved categories. We find that common and visually salient abnormalities (e.g., organ enlargement, soft-tissue collections) are generally solved reliably, whereas fine-grained or anatomically localized find-

ings—particularly musculoskeletal pathologies—remain difficult for all models. Although medically oriented models perform competitively on frequent categories, they do not consistently outperform general-purpose models on the harder ones, and InternVL often achieves the best results overall, suggesting that detailed disease characterization rather than abnormality detection constitutes the main bottleneck.

### C.3. Robustness to question template variations

To assess whether models exploit linguistic shortcuts or are overly sensitive to question phrasing, we analyze model performance across different natural-language templates within the same task, category, and question type. Each template corresponds to a distinct formulation of an equivalent semantic query (e.g., abnormality detection or pathology presence).

Table 8 reports accuracy for each model across all templates. Templates are well distributed within each task category, and no single phrasing dominates the benchmark. We note that different templates are instantiated on different image subsets; therefore, small variations in accuracy may partially reflect sample variability rather than purely linguistic effects.

Overall, model performance remains relatively stable across paraphrased templates within the same task, suggesting that the benchmark does not contain strong template-specific shortcuts. This is particularly evident in anatomical region and structure identification tasks, where accuracy varies only marginally across formulations. However, higher sensitivity is observed for certain models in linguistically challenging settings, notably open-ended pathology identification and closed-form pathology detection on negative cases. For example, in open-ended pathology identification, MedGemma varies from 26.28% to 34.70% across paraphrased templates, while InternVL3.5 drops from 14.99% to 8.58%. In negative pathology detection, InternVL3.5 ranges from 48.39% to 7.71% and GPT-5 from 73.66% to 8.44% depending on phrasing, whereas MedGemma and Lingshu remain comparatively stable (typically >75%). These variations indicate model-specific sensitivity to wording rather than systematic bias in the benchmark templates.

Taken together, these results support the robustness of our question templating strategy while revealing meaningful differences in how models handle linguistic variation.

### C.4. Out-of-template evaluation of fine-tuned models

While RadImageNet-VQA provides carefully designed question templates to reduce superficial language biases, these templated formats may inadvertently encourage models to rely on repeated linguistic patterns rather than true visual understanding. To assess the robustness of our fine-tuned models beyond these constraints, we conducted an out-of-template evaluation, in which questions were rephrased or presented in styles not seen during training. This setup simulates realistic clinical queries, including open-ended descriptions and higher-level reasoning, that go beyond the template structures seen in training (e.g., asking "What do you see in the heart?" instead of "Which body part is visible in this image?").

For this experiment, we tested three representative CT/MRI images: a shoulder scan with soft-tissue fluid, a spine scan with foraminal pathology, and an abdominal scan with uterine pathology. Questions were designed to probe both anatomical identification and

Table 8: **Per-template accuracy (%)** across models, grouped by question type (Closed+/Closed-/Open-ended/MCQ). Count denotes the number of benchmark questions instantiating each template.

| Question Template | Count | InternVL | Qwen-VL | GPT-5 | MedGemma | Lingshu |
|---|---|---|---|---|---|---|
| *Abnormality Detection - Closed* | | | | | | |
| Does the {anatomy} look abnormal? | 266 | 77.07 | 72.18 | 25.94 | 60.90 | 50.38 |
| Does this scan reveal any abnormalities? | 220 | 76.36 | 58.64 | 18.18 | 56.36 | 39.55 |
| Is there an abnormality present? | 256 | 73.05 | 72.27 | 44.53 | 55.86 | 51.95 |
| Is this image abnormal? | 258 | 71.32 | 73.26 | 20.16 | 48.45 | 48.45 |
| *Pathology Identification - Open-ended* | | | | | | |
| What abnormality is present? | 487 | 14.99 | 10.47 | 18.69 | 26.28 | 15.61 |
| What condition is affecting the anatomy? | 513 | 8.58 | 9.16 | 13.06 | 34.70 | 15.79 |
| *Pathology Detection - Closed (Positive Cases)* | | | | | | |
| Are there signs of {pathology}? | 196 | 84.69 | 65.82 | 51.02 | 48.98 | 54.08 |
| Does the patient have {pathology}? | 195 | 87.18 | 69.23 | 57.44 | 58.97 | 65.64 |
| Is the {anatomy} affected by {pathology}? | 190 | 88.42 | 67.89 | 54.21 | 53.16 | 58.42 |
| Is there evidence of {pathology} in this image? | 218 | 86.24 | 73.39 | 53.21 | 52.29 | 51.38 |
| Is {pathology} present? | 201 | 87.06 | 69.15 | 50.75 | 57.71 | 56.22 |
| *Pathology Detection - Closed (Negative Cases)* | | | | | | |
| Are there signs of {pathology}? | 186 | 48.39 | 53.23 | 73.66 | 81.72 | 84.95 |
| Does the patient have {pathology}? | 206 | 22.33 | 38.83 | 59.71 | 67.96 | 66.02 |
| Is the {anatomy} affected by {pathology}? | 200 | 18.50 | 52.00 | 61.00 | 77.00 | 80.00 |
| Is there evidence of {pathology} in this image? | 218 | 7.71 | 42.20 | 8.44 | 84.86 | 83.03 |
| Is {pathology} present? | 190 | 31.58 | 52.11 | 68.42 | 75.26 | 80.53 |
| *Pathology Classification - Multiple Choice* | | | | | | |
| Select the pathology that best describes this image | 219 | 47.03 | 27.85 | 42.92 | 31.05 | 25.11 |
| What is the most likely pathology in this image? | 260 | 46.15 | 31.92 | 40.77 | 36.54 | 29.62 |
| What pathology is shown in this scan of {anatomy}? | 250 | 48.80 | 32.40 | 39.60 | 36.80 | 28.80 |
| Which of the following pathologies is present in this image? | 271 | 46.49 | 28.04 | 41.70 | 41.70 | 33.95 |
| *Anatomical Region Identification - Open-ended* | | | | | | |
| Name the organ featured in this image | 177 | 49.72 | 25.99 | 36.16 | 54.24 | 55.93 |
| What anatomical region is shown? | 162 | 65.43 | 46.30 | 58.64 | 69.75 | 54.94 |
| What organ is highlighted in this scan? | 183 | 53.01 | 28.42 | 34.97 | 66.12 | 45.36 |
| What specific anatomical structure is highlighted? | 172 | 39.53 | 22.09 | 8.14 | 51.74 | 30.81 |
| Which body part is visible in this image? | 156 | 68.59 | 58.97 | 64.10 | 75.00 | 60.26 |
| Which part of the body does this image belong to? | 150 | 66.67 | 48.00 | 70.67 | 62.00 | 52.00 |
| *Anatomical Structure Presence Detection - Closed* | | | | | | |
| Can you identify the {anatomy} in this image? | 146 | 82.18 | 80.77 | 30.94 | 48.82 | 86.63 |
| Can you see the {anatomy}? | 136 | 88.49 | 85.75 | 86.96 | 85.84 | 90.33 |
| Does the picture contain the {anatomy}? | 139 | 88.96 | 81.73 | 86.43 | 84.81 | 88.79 |
| Does this image show the {anatomy}? | 145 | 88.38 | 80.62 | 82.92 | 87.30 | 86.83 |
| Is the {anatomy} part of this scan? | 139 | 84.69 | 79.94 | 80.28 | 80.83 | 85.89 |
| Is the {anatomy} present in this image? | 145 | 87.46 | 84.84 | 85.84 | 84.83 | 89.70 |
| Is the {anatomy} visible in the image? | 152 | 83.78 | 80.13 | 87.18 | 83.90 | 86.94 |
| *Anatomical Region Selection - Multiple Choice* | | | | | | |
| Select the correct anatomical region | 342 | 88.30 | 79.82 | 88.60 | 83.33 | 87.13 |
| What is the correct anatomical region for this scan? | 321 | 88.79 | 79.44 | 88.47 | 82.87 | 89.10 |
| Which of the following anatomical regions is depicted in this scan? | 337 | 92.58 | 82.20 | 90.80 | 88.13 | 90.50 |

pathology interpretation, including general prompts, localization questions, and complication reasoning. The qualitative results are summarized in Figure 12.

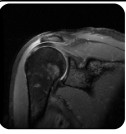

Q: **Explain what is abnormal, if anything.**

**LLaVA-OV-FT**: labral pathology
**Qwen2.5-VL-FT**: soft tissue fluid
**Lingshu-FT**: soft tissue fluid

Q: **What do you see on the left side of the image?**

**LLaVA-OV-FT**: On the left side of the image, there is a subacromial cyst. A subacromial cyst is a fluid-filled sac that forms in the subacromial space, which is located beneath the acromion, a bony projection on the scapula (shoulder blade).
**Qwen2.5-VL-FT**: On the left side of the image, there is a fluid-filled subacromial bursa. The subacromial bursa is a small, fluid-filled sac that reduces friction between the shoulder joint and the muscles, tendons, and bones surrounding it.
**Lingshu-FT**: On the left side of the image, there is a fluid collection visible. This could be indicative of inflammation, injury, or other pathological changes in the area.

Q: **Write a radiologist-style report of this scan.**

**LLaVA-OV-FT**: This MRI scan showsfacet arthropathy with degenerative changes at the L4-L5 level, which is consistent with degenerative disc disease. Additionally, there is a right L4-L5 foraminal herniated disc with mild impingement of the right L5 nerve root.
**Qwen2.5-VL-FT**: This axial T2-weighted MR image demonstrates a foraminotomy at the L4-5 level with a small residual disc protrusion and mild right foraminal stenosis. There is no significant nerve root compression.
**Lingshu-FT**: Findings: This axial T2-weighted MRI sequence demonstrates the cervical spine with 1 abnormality identified. There is a C5-6 disc protrusion, which is causing spinal cord compression and resulting in the effacement of the spinal cord signal. This finding is consistent with spinal cord compression at the C5-6 level. Impression: C5-6 disc protrusion with spinal cord compression.

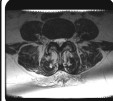

Q: **Focus on the inferior position - what is present ?**

**LLaVA-OV-FT**: On the left side of the image, there is a large disc protrusion at the L4-L5 level causing narrowing of the left lateral recess and compression of the left nerve root.
**Qwen2.5-VL-FT**: On the left side of the image, there is a disc protrusion.
**Lingshu-FT**: On the left side of the image, there is a disc protrusion at the L4-L5 level, which is causing compression of the thecal sac and left nerve root.

Q: **Identify the liver in this shoulder image.**

**LLaVA-OV-FT**: spine
**Qwen2.5-VL-FT**: There is no liver present in this image as it is an MRI of the spine, not the shoulder or liver region.
**Lingshu-FT**: There is no liver visible in this shoulder image.

Q: **Which structure occupies the posterior part of the image?**

**LLaVA-OV-FT**: dural or epidural tissue
**Qwen2.5-VL-FT**: The spinal cord occupies the posterior part of the image.
**Lingshu-FT**: spine

Q: **What possible complications could arise from this abnormality?**

**LLaVA-OV-FT**: dural or epidural abscesses can lead to serious complications, such as nerve compression, neurological deficits, or even paralysis.
**Qwen2.5-VL-FT**: Detailed 4-sentence paragraph discussing nerve compression, pain, instability, degeneration
**Lingshu-FT**: dural or epidural hematoma, spinal cord injury, worsening of the fracture, or infection

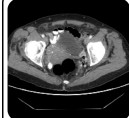

Q: **Identify the liver in this shoulder image.**

**LLaVA-OV-FT**: There is no liver visible in this image as it is a CT scan of the colon and right kidney
**Qwen2.5-VL-FT**: There is no liver visible in this image as it is a pelvic CT scan
**Lingshu-FT**: supraspinatus

Q: **Are there any high-signal regions?**

**LLaVA-OV-FT**:This CT scan shows the kidneys, which are a pair of bean-shaped organs in the human body responsible for filtering waste products, excess water, and other impurities from the blood.
**Qwen2.5-VL-FT**: There are none.
**Lingshu-FT**: No

Q: **What is visible in the upper part of the scan?**

**LLaVA-OV-FT**: In the upper part of the CT scan, there is a large mass visible. This mass appears to be a solid mass with a high density, which is indicated by the white arrow.
**Qwen2.5-VL-FT**: In the upper part of the scan, the rectal wall is visible.
**Lingshu-FT**: In the upper part of the scan, there is a large amount of ascites, which is an abnormal accumulation of fluid in the abdominal cavity.

Q: **Focus on the inferior portion — what is present?**

**LLaVA-OV-FT**: ascites
**Qwen2.5-VL-FT**: ascites
**Lingshu-FT**: ascites

Q: **What possible complications could arise from this abnormality?**

**LLaVA-OV-FT**:An abnormality like a large pelvic mass, as seen in the CT scan, can lead to various complications. Some possible complications include:
1. Compression or displacement of nearby organs or structures, which may affect their function.
2.obstruction or strangulation of blood vessels or nerves, which can lead to reduced blood flow or nerve function.
3. Infection or inflammation, which may cause additional complications if not addressed.
4. Tumor necrosis or hemorrhage, which can lead to tissue death or bleeding.
5. Metastasis, which is the spread of cancer cells from the primary tumor to other parts of the body.
It is important to consult a healthcare professional for a thorough evaluation and proper diagnosis of the underlying cause of the abnormality, as well as to determine the most appropriate treatment and management plan.
**Qwen2.5-VL-FT**: The abnormality in the image, which shows a dilated appendix with a diameter of 10 mm, could lead to several complications if left untreated. These may include:
1. Acute appendicitis: This is the most common complication, where the appendix becomes inflamed and swollen due to an infection or obstruction. Symptoms may include severe abdominal pain, fever, nausea, and vomiting.
2. Abscess formation: In some cases, the infection may spread and cause the formation of an abscess, which is a collection of pus within the tissue. This can lead to more severe symptoms and may require surgical intervention.
3. Peritonitis: If the infection spreads beyond the appendix, it can cause inflammation of the peritoneum, the thin layer of tissue that lines the abdominal cavity. This can lead to severe abdominal pain, fever, and other symptoms.
4. Fistula formation: In rare cases, the appendix may develop a fistula, which is an abnormal connection between the appendix and another nearby structure, such as the bladder or intestine.
It is important to consult a healthcare professional for a thorough evaluation and proper diagnosis of the underlying cause of the abnormality. They can recommend appropriate treatment options to prevent or manage these complications.
**Lingshu-FT**: Complications may include infection, rupture, or obstruction depending on the nature and extent of the mass.

Figure 12: Out-of-template evaluation of three fine-tuned VLMs (LLaVA-OneVision-FT, Qwen2.5-VL-FT, Lingshu-FT) on representative CT/MRI scans. Questions are phrased in ways not seen during training, including open-ended descriptions and higher-level reasoning queries. Cells summarize whether the model correctly identifies anatomy/pathology, provides plausible but partially incorrect answers, or hallucinates information.

Overall, the models demonstrate several encouraging behaviors. All three fine-tuned VLMs are able to correctly identify visible anatomical structures in many cases, even when questions are phrased differently from the training templates. This indicates that the models have learned meaningful visual representations and can generalize their anatomical knowledge beyond templated cues. Additionally, Qwen2.5-VL-FT and Lingshu-FT often provide clinically plausible explanations for observed abnormalities, showing that they are capable of reasoning about pathology in a way that is consistent with image content. In particular, these models handle absent structures conservatively, avoiding hallucination when organs or pathologies are not present, which reflects a careful grounding in the visual evidence. In some instances, LLaVA-OV-FT generates detailed stepwise clinical descriptions, demonstrating the potential for synthesizing information into structured, radiologist-style explanations.

The evaluation also highlights areas where improvements are needed. LLaVA-OV-FT occasionally hallucinates structures or pathologies in response to out-of-template questions, indicating reliance on linguistic patterns learned during training. Complex pathology questions, such as those involving potential complications or higher-level interpretations, are more prone to over-specification or partially incorrect reasoning, particularly for models heavily influenced by template patterns. These observations suggest that while current fine-tuned models can generalize beyond template formats, their robustness is limited when faced with more diverse or clinically nuanced question phrasing.

These findings underscore the importance of expanding the diversity of training data, both in the way questions and answers are formulated and in the structure of multi-turn interactions. By incorporating multiple paraphrases, open-ended reasoning prompts, and varied conversational styles, future work can further enhance the generalization capacity of vision-language models in radiologic VQA, reducing template dependency and improving reliability in real-world clinical settings.

## C.5. Extended results on text-only evaluation

Figure 4 shows that text-only performance on RadImageNet-VQA drops close to the random baseline, suggesting that most multiple-choice questions cannot be solved without visual evidence. To better understand the residual accuracy that remains, Figure 13 breaks results down by abnormality status. When the image is removed, medical-oriented models such as MedGemma-4B almost always choose the "no pathology seen" option, which yields very high accuracy for normal cases but causes performance on abnormal cases to collapse. In contrast, general-purpose models (GPT-5 and InternVL3.5-14B) do not rely on this conservative default and instead spread predictions across all choices, resulting in lower but more

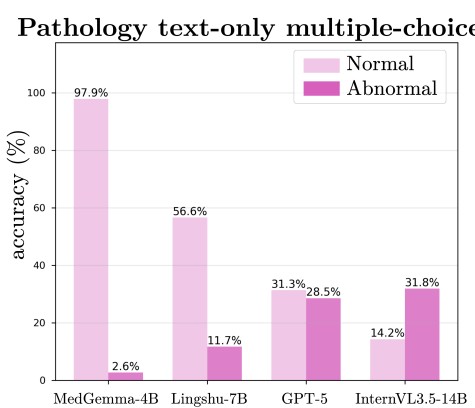

Figure 13: MC text-only accuracy on RadImageNet-VQA.

balanced scores. These results show that the small amount of text-only accuracy observed on RadImageNet-VQA primarily reflects conservative guessing strategies rather than the ability to infer pathology from the question text alone, further confirming that pathology identification requires image grounding.

## C.6. More qualitative examples

The examples in Figure 14 illustrate how VQA-RAD and SLAKE contain strong textual cues that enable models to answer correctly even without visual input. In each case, the model produces essentially the same response whether it receives the image and question or the question alone, showing that the question text often provides enough information to determine the answer. For instance, in *Where does the left renal vein connect to?*, the mention of *renal vein* directly implies the *inferior vena cava*, a standard anatomical relationship independent of the specific CT slice. Likewise, questions such as *Which organ is part of the urinary system?* or *Which organs are part of the digestive system?* contain the key semantic identifiers—*urinary system*, *digestive system*—that uniquely determine the correct organ without requiring visual confirmation. Even pathology-oriented prompts (e.g., *What disease is shown on the right lung?*) can be shortcut through frequent-label priors, leading the model to answer *pneumonia* regardless of the underlying image.

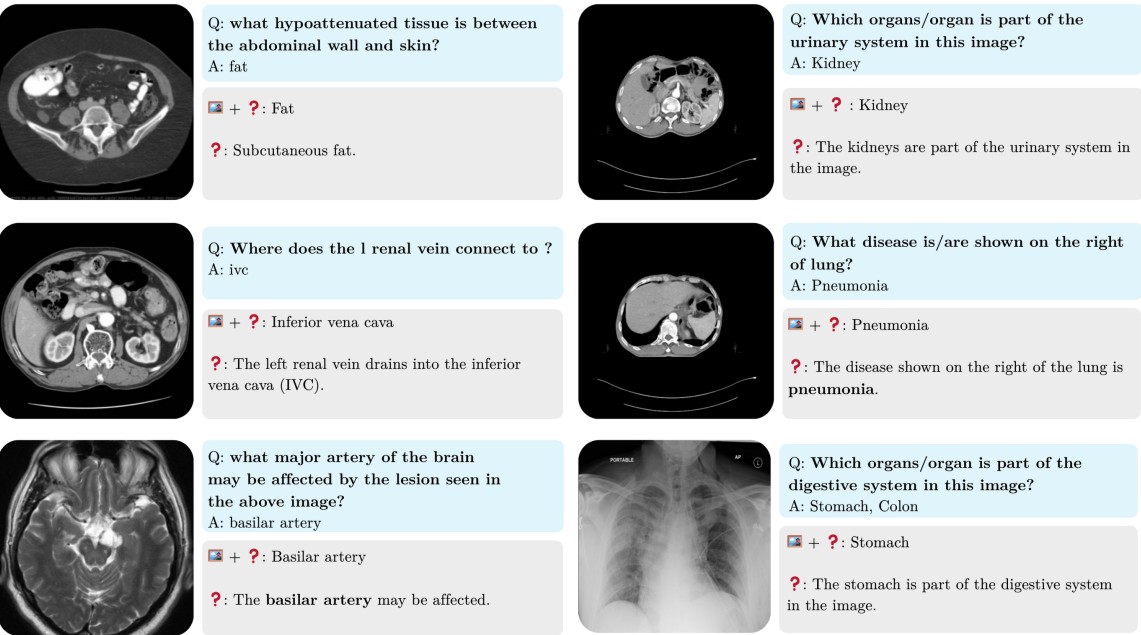

Figure 14: Examples of samples from VQA-RAD and SLAKE where current models do not need the image. The responses were obtained with MedGemma-4B.

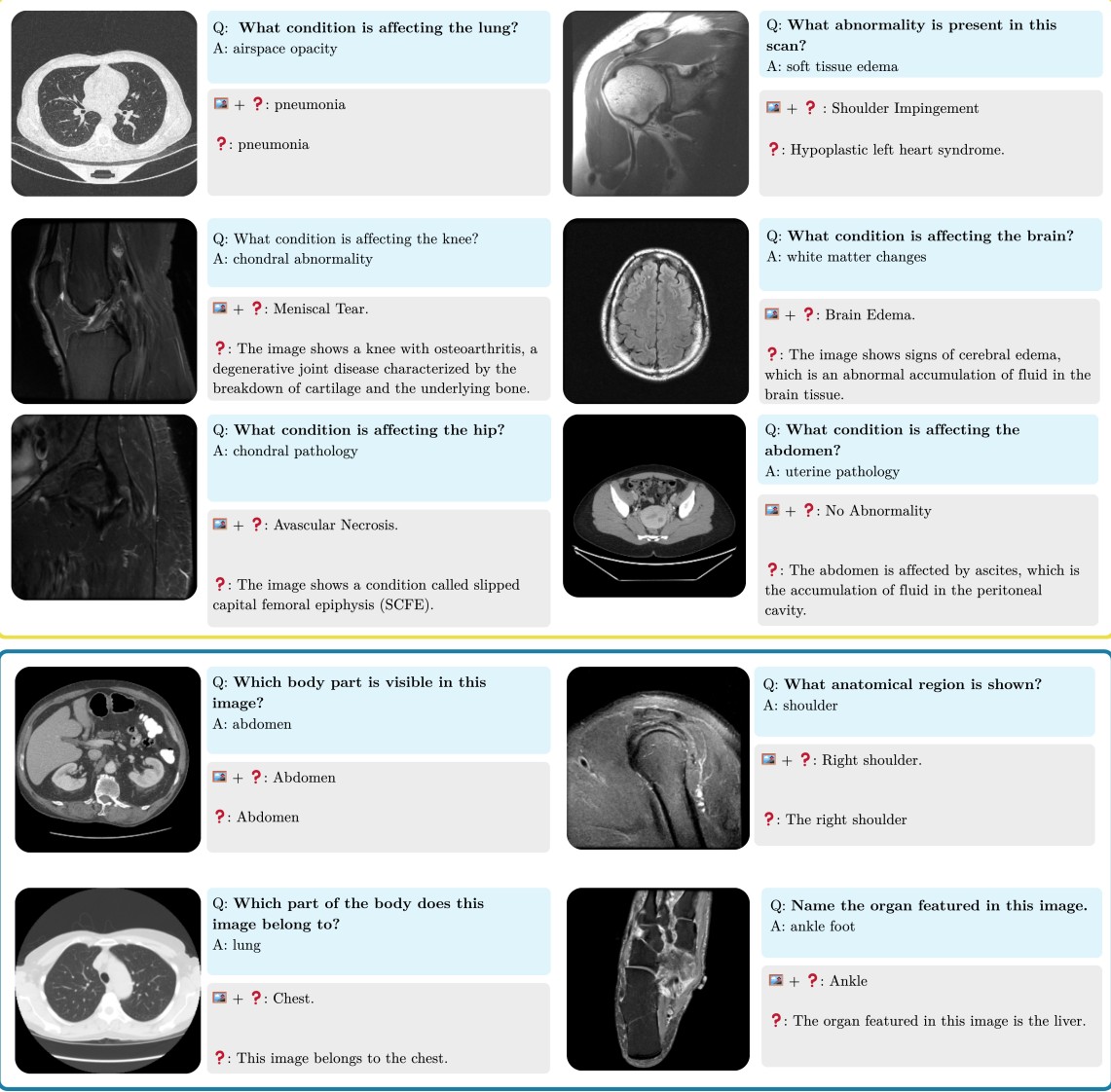

Figure 15: Examples comparing Lingshu-7B's answers when given the image and question versus the question alone on RadImageNet-VQA.

Building on the text-only results in Figure 4, where RadImageNet-VQA yields much lower shortcutability than VQA-RAD or SLAKE but where Lingshu-7B still attains 9.4% accuracy on open-ended questions, we further inspect its predictions with and without images. Figure 15 shows qualitative examples on RadImageNet-VQA for both pathology and anatomy questions. When the prompt explicitly mentions an organ or region (e.g., lung, knee, hip, brain), the text-only model often maps this cue to a high-frequency pathology for that location—such as *pneumonia* for the lung or *osteoarthritis* for the knee—which sometimes coincides with the image-based answer (e.g., *pneumonia*, *cerebral/brain edema*)

but often diverges from it (e.g., *osteoarthritis* vs. *meniscal tear, ascites* vs. no abnormality, *SCFE* vs. *avascular necrosis*). This behavior indicates that, in the absence of visual evidence, Lingshu falls back on learned pathology priors tied to anatomical templates rather than genuinely reasoning about the specific case. For anatomy questions, multimodal predictions are typically correct and well localized (abdomen, chest, shoulder, ankle), whereas text-only outputs tend to overuse very frequent regions such as abdomen, chest, or liver, reflecting the underlying dataset distribution more than the individual question. Overall, these examples suggest that RadImageNet-VQA still leaves a narrow band of text-based shortcuts when anatomical regions are explicitly named, but that many failures in the multimodal setting stem from the intrinsic difficulty of fine-grained pathology recognition and from the model's reliance on organ and frequency-based priors under limited cues, rather than from purely linguistic shortcutting alone.

## Appendix D. Extended Fine-Tuning Details

### D.1. Training data composition

To train radiologic VLMs, we assemble a multimodal corpus composed of: (1) RadImageNet-VQA train split. (2) additional CT/MRI converted into 2D VQA pairs KiTS22 (Cardoso et al., 2017), and AbdomenAtlas (Chen et al., 2025a), where 3D volumes are sliced into 2D images and paired with captions or VQA questions derived from organ and lesion annotations; and (3) existing radiologic VQA datasets such as VQA-RAD (Lau et al., 2018), SLAKE (Liu et al., 2021), and radiology subset of LLaVA-Med (Li et al., 2023a).

| Dataset | # Images | Alignment | Instruction |
|---|---|---|---|
| RadImageNet (Mei et al., 2022) | 750K | 750K | 6.75M |
| KiTS22 (Cardoso et al., 2017) | 4.6K | 4.6K | 32.5K |
| AbdomenAtlas (Chen et al., 2025a) | 43.7K | 43.7K | 211K |
| LLaVA-Med (Li et al., 2023a) | 38.4K | 13.8K | 72.3K |
| VQA-RAD (Lau et al., 2018) | 0.3K | – | 1.8K |
| SLAKE (Liu et al., 2021) | 0.4K | – | 4.9K |
| **Total** | **837.5K** | **812.1K** | **7.07M** |

Table 9: Summary of datasets used for alignment and instruction tuning, including the number of images and approximate QA pair counts.

Together, these three branches feed into a unified training corpus comprising medical instruction data and medical alignment data, as depicted in Figure 16.

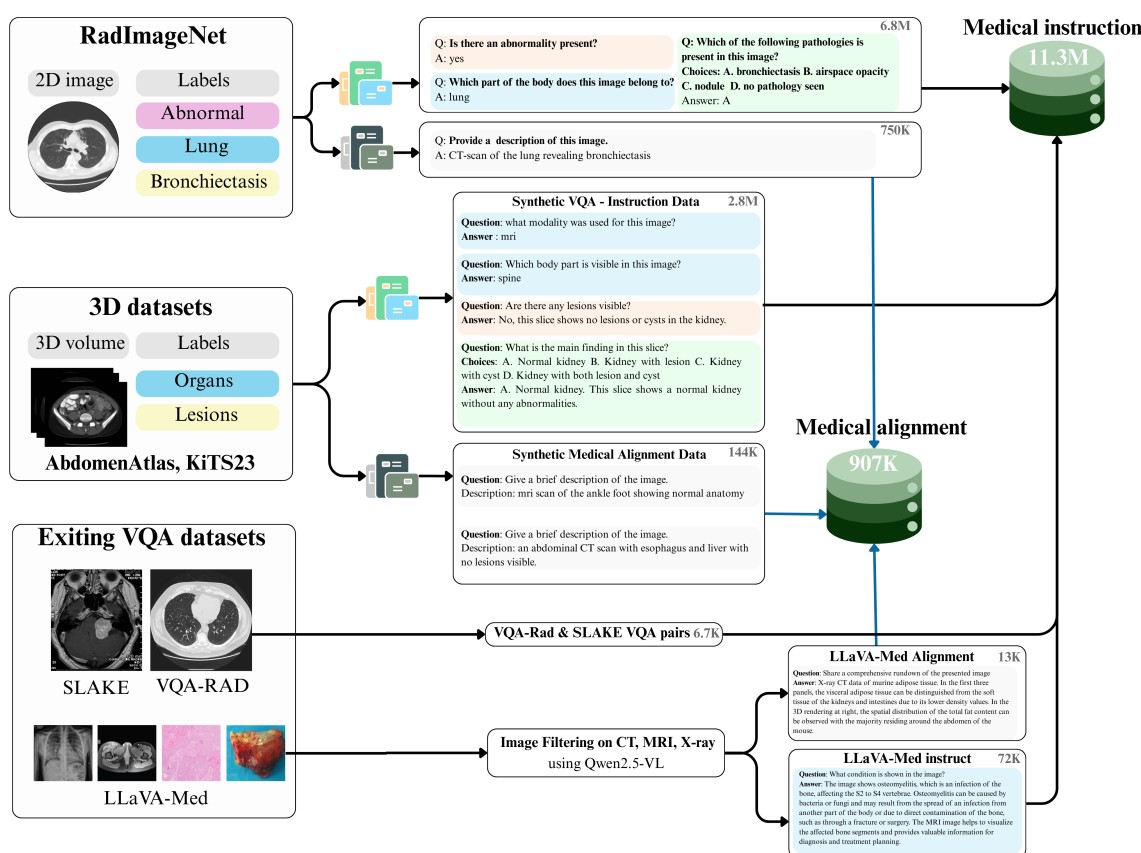

Figure 16: The overall data curation pipeline of radiologic multimodal data

## D.2. Implementation Details

| Category | Configuration |
|---|---|
| **Training stages** | Visual alignment (1 epoch) |
| | Instruction tuning (2 epochs) |
| **Batching and scheduling** | Global batch size: 16 (4 GPUs, per-GPU batch = 1, gradient accumulation = 4) |
| | Learning rate: 1e–5 (LLM), 2e–6 (vision tower) |
| | Warm-up: 3% of total steps |
| | LR schedule: cosine decay |
| **Optimization** | Optimizer: AdamW ($\beta_1 = 0.9$, $\beta_2 = 0.999$) |
| | Weight decay: 0.0 |
| **Evaluation parameters** | Max new tokens: 8192 |
| | Decoding: temperature = 0, top-p = 0.0001 |
| | Repetition penalty: 1.0 |
| | Judge backend: `mistral-large-latest` |
| | Random seed: 42. |

Table 10: Training and evaluation configuration for RadImageNet-VQA experiments.

We fine-tune all models described in Section 4.3 with 4x **NVIDIA H100 GPUs**. Training follows a two-stage procedure: (1) a *visual alignment* phase in which only the vision tower and projector are updated while the language model remains frozen, and (2) an *instruction tuning* phase where all components are trainable. Evaluation uses a unified decoding configuration and a fixed LLM-as-a-judge setup for all models. The full set of hyperparameters and evaluation settings is summarized in Table 10.

### D.3. Full fine-tuning results

Table 11 reports the exact fine-tuned values and deltas for each task and question type on RadImageNet-VQA. This show that all models benefit substantially from radiologic instruction tuning, with very large gains on abnormality and open-ended pathology questions and near-ceiling performance on anatomy.

| Task | LLaVA-One-Vision | Lingshu-7B | QwenVL-2.5-7B |
|---|---|---|---|
| **Anatomy** | | | |
| Open | 99.4 ↑(+51.0) | 99.4 ↑(+49.8) | 99.3 ↑(+61.8) |
| Closed+ | 87.4 ↑(+4.7) | 95.2 ↑(+4.5) | 93.4 ↑(+8.5) |
| Closed– | 97.8 ↑(+16.5) | 97.6 ↑(+12.5) | 97.3 ↑(+18.2) |
| MC | 99.4 ↑(+10.7) | 98.9 ↑(+10.0) | 98.5 ↑(+18.0) |
| **Abnormality** | | | |
| Closed | 87.6 ↑(+37.8) | 87.7 ↑(+39.8) | 85.9 ↑(+16.4) |
| **Pathology** | | | |
| Open | 42.6 ↑(+26.6) | 41.2 ↑(+25.5) | 39.2 ↑(+29.4) |
| Closed+ | 67.2 ↑(+11.9) | 84.3 ↑(+27.3) | 74.7 ↑(+5.5) |
| Closed– | 77.2 ↑(+15.9) | 65.3 ↓(-13.5) | 68.1 ↑(+20.7) |
| MC | 60.1 ↑(+26.5) | 49.9 ↑(+20.3) | 53.4 ↑(+23.3) |
| **Avg** | 79.9 ↑(+22.4) | 79.9 ↑(+19.5) | 78.9 ↑(+22.5) |

Table 11: Fine-tuned performance on RadImageNet-VQA with delta values relative to zero-shot baselines. Improvements are shown in green and regressions in red.

