# OpenReview forum: "RadImageNet-VQA: A Large-Scale CT and MRI Dataset for Radiologic Visual Question Answering"
_MIDL.io/2026/Conference — MIDL 2026 Poster_

### Official Review · Reviewer_dPjo · 2025-12-21

**Confidence:** 4
**Preliminary Rating:** 4

**Summary:**

This paper introduces RadImageNet-VQA, a large-scale radiologic visual question answering dataset built from expert-annotated CT and MRI images. The dataset covers anatomy recognition, abnormality detection, and fine-grained pathology identification across multiple question formats, providing 750K images paired with 7.5M QA samples. Through extensive zero-shot and fine-tuning experiments on state-of-the-art vision–language models, the authors show that anatomy recognition is largely saturated, whereas pathology identification remains challenging, particularly in open-ended settings. Text-only evaluations demonstrate that the dataset substantially reduces linguistic shortcuts, supporting its validity as a diagnostic benchmark. Overall, the work positions RadImageNet-VQA as a scalable resource for training and evaluating radiologic VQA models.

**Strengths:**

1. The dataset significantly expands prior medical VQA resources by offering unprecedented scale for CT and MRI, two modalities underrepresented in existing benchmarks. The combination of 750K images and 7.5M QA samples provides strong potential for both evaluation and instruction tuning of vision–language models.

2. The decomposition into anatomy recognition, abnormality detection, and pathology identification closely reflects how radiologists interrogate imaging data. The inclusion of multiple question formats further enables nuanced analysis of model capabilities under different reasoning demands.

3. The paper provides convincing evidence, through text-only ablations and distractor design analysis, that RadImageNet-VQA substantially reduces linguistic shortcuts compared to prior datasets. This directly addresses a known weakness of medical VQA benchmarks and strengthens the validity of the evaluation.

4. The authors benchmark a wide range of both general-purpose and medical-oriented VLMs under zero-shot and fine-tuned settings. The analysis is thorough, covering task-level performance, question-type breakdowns, text-only baselines, and ablations on vision encoders and data sampling strategies.

5. Findings such as the limited benefit of medically pretrained vision encoders and the persistent difficulty of fine-grained pathology identification provide valuable guidance for future model and dataset design, beyond the immediate contribution of the dataset itself.

**Weaknesses:**

1. Although the authors design multiple templates per task, the QA pairs remain programmatically generated. This may restrict linguistic variability and reasoning richness compared to human-authored questions, potentially limiting transfer to more open-ended clinical dialogue settings.

2. The correctness of free-text responses is assessed using an automatic judge model, which introduces an additional layer of uncertainty. While practical, this choice may conflate model errors with judge bias, and its reliability for fine-grained pathology distinctions is not deeply validated.

3. The curated evaluation benchmark contains only 1,000 images and 9,000 QA pairs. Given the scale of the dataset, this raises questions about statistical robustness and whether the benchmark fully captures the long-tail diversity of pathologies and anatomies present in the training set.

4. The dataset formulation assumes a single ground-truth pathology label per image, whereas real radiologic cases often involve multiple co-existing findings or diagnostic uncertainty. This simplification may constrain the dataset’s fidelity to real-world clinical interpretation.

5. Although motivated as complementary to emerging 3D VLMs, the dataset is strictly 2D. The paper does not empirically analyze how well conclusions drawn from RadImageNet-VQA transfer to volumetric or longitudinal radiology scenarios.

**Detailed Comments:**

1. Clarifying how often multiple pathologies co-occur in the source RadImageNet labels, and how such cases are handled during QA generation, would improve transparency.

2. Additional quantitative analysis of inter-template variability, such as performance stratified by template families, could strengthen claims about shortcut mitigation.

3. The paper would benefit from a brief discussion on ethical considerations and potential dataset biases inherited from RadImageNet, especially given the dataset’s intended public release.

**Justification Of The Preliminary Rating:**

The paper makes a meaningful contribution by releasing a large-scale, CT/MRI-focused VQA dataset that addresses key limitations of prior benchmarks, particularly shortcut bias and limited modality coverage. While the approach relies on template-based QA generation and exhibits moderate novelty, the dataset scale, careful construction, and thorough empirical analysis provide clear value to the community. The experimental results yield useful insights into current model limitations and training strategies for radiologic VQA. Despite some concerns regarding clinical realism and evaluation methodology, the overall contribution is solid and merits acceptance, albeit with a narrow margin, supporting a weak accept recommendation.

**Questions To Address In The Rebuttal:**

1. How robust is the LLM-as-a-judge evaluation for fine-grained pathology identification, and were alternative validation strategies considered?

2. Can the authors comment on how representative the 1,000-image benchmark subset is with respect to rare pathologies and anatomical regions in the full dataset?

3. To what extent do the authors expect models trained on RadImageNet-VQA to generalize to truly free-form clinical questions beyond the predefined templates?

---

> ### Author Response · Authors · 2026-01-24
> **Response to Reviewer dPjo**
>
> We thank the reviewer for the thoughtful feedback. We are encouraged by your recognition of the dataset’s scale, the clinical relevance of the task decomposition, and the breadth of the experimental evaluation. We hope the following responses address your concerns.
>
> &nbsp;
>
> > **How robust is the LLM-as-a-judge evaluation for fine-grained pathology identification, and were alternative validation strategies considered?**
>
> We use an LLM-as-a-judge (Mistral-Large 2.1) for open-ended answers specifically to account for semantic equivalence and to avoid penalizing clinically correct paraphrases. In response to this comment, we conducted a manual evaluation of a representative subset of 400 open-ended predictions sampled across tasks and models, and scored by human annotators. The LLM judge matches human evaluation in 90.6% of cases, achieving a substantial agreement with a Cohen’s κ score of 0.72, supporting the reliable assessment across different model linguistic styles. We include the protocol and results in the revised manuscript (Section 3.2 and Appendix B.2).
>
> &nbsp;
>
> > **Can the authors comment on how representative the 1,000-image benchmark subset is with respect to rare pathologies and anatomical regions in the full dataset?**
>
> The 1,000-image benchmark is built via stratified sampling from the original test split of RadImageNet to preserve the distribution across the 8 anatomical regions and 97 pathology categories. In the revised manuscript, we add per-category counts and frequencies for anatomy and pathology in both the benchmark and the training corpus (Appendix A.2). The benchmark distribution reflects the training corpus; for instance, the abdomen represents 31.3% of the benchmark images versus 25.2% in training, while the lung accounts for 19.3% versus 12.0% in training, showing consistent relative prevalence. For rare pathologies (<1% prevalence), we slightly upsample them in the benchmark to ensure sufficient instances for stable evaluation. For example, “pancreatic lesion” constitutes 0.35% of the training set but is represented in 0.7% of the benchmark images, and “spring ligament injury” is 0.01% in training versus 0.3% in the benchmark.
>
> &nbsp;
>
> > **To what extent do the authors expect models trained on RadImageNet-VQA to generalize to truly free-form clinical questions beyond the predefined templates?**
>
> > **Additional quantitative analysis of inter-template variability, such as performance stratified by template families, could strengthen claims about shortcut mitigation.**
>
> Following the reviewer's suggestion, we add two analyses in the revised manuscript. First, we evaluate fine-tuned models on out-of-template questions created via manual edits and never seen during training (e.g., laterality “right/left kidney”) (Appendix C.4). Qualitative results show models perform well on these unseen phrasing variations, suggesting they learn visual–textual medical concepts rather than memorizing template strings. Second, we report a template-stratified performance breakdown (Appendix C.3) to quantify inter-template variability. Performance is largely stable across template families, providing evidence that results are not driven by specific linguistic formulations.
>
> &nbsp;
>
> > **Clarifying how often multiple pathologies co-occur in the source RadImageNet labels, and how such cases are handled during QA generation, would improve transparency.**
>
> RadImageNet-VQA inherits the image-level labeling scheme of RadImageNet, which provides a single pathology label per image. We agree that many real-world clinical cases involve multiple co-existing findings. To improve transparency, in the revised manuscript we add the single-label constraint as a limitation (Section 5). Extending supervision to co-existing findings is an important direction for future work.
>
> &nbsp;
>
> > **The paper would benefit from a brief discussion on ethical considerations and potential dataset biases inherited from RadImageNet, especially given the dataset’s intended public release.**
>
> In the revised manuscript, we add a dedicated section (Section 5) discussing ethical considerations and potential biases inherited from the RadImageNet source. We acknowledge the single-label annotation constraint, biases in pathology prevalence and anatomical coverage, and the limitations these impose for real-world clinical deployment.

---

### Official Review · Reviewer_WnEX · 2025-12-31

**Confidence:** 4
**Preliminary Rating:** 5
**Final Rating:** 5

**Summary:**

This paper introduces RadImageNet-VQA, a newly compiled radiologic visual question answering (VQA) dataset focused on CT and MRI images. The authors build upon the RadImageNet CT/MRI collection, leveraging its expert-annotated anatomy and pathology labels to generate over 750K images paired with 7.5M caption/Q-A samples. The dataset includes three task families: anatomy recognition, abnormality detection, and fine-grained pathology identification. It spans 8 anatomical regions and 97 pathology categories, featuring open-ended, yes/no (closed), and multiple-choice questions.

**Strengths:**

1. The dataset’s size and diversity are truly unprecedented for radiologic and is the largest CT/MRI coverage and the most extensive QA corpus compared to existing datasets.
2. By focusing on CT/MRI (rather than X-ray or abstract biomedical images) and by providing an order-of-magnitude larger dataset, this work has the potential to have a high impact on medical VQA research.
3. The experimental work is extensive. The authors benchmark a wide range of VLMs (open-source and proprietary, general-purpose and medical-specialized) under zero-shot and fine-tuning conditions.
4. The manuscript is well-structured.

**Weaknesses:**

The Q-A pairs are generated via prompt templates using RadImageNet metadata. While this scales easily, there is a risk that the questions (and possibly answers) have repetitive or unnatural phrasing. The paper provides some examples of templates (Fig. 7) and counts (Table 3), but it is unclear how linguistically diverse the resulting questions are.

**Detailed Comments:**

This comprehensive open-source dataset provides a foundation for medical VQA research, and this paper is well-organized and well-written.

**Justification Of Final Rating:**

Thanks for the response. The authors have addressed my concerns and comments regarding the potential repetitiveness of template-generated questions. I will maintain my score for this paper of 5: Strong accept.

**Justification Of The Preliminary Rating:**

RadImageNet-VQA represents a substantial and well-executed contribution to medical imaging and multimodal learning. The dataset’s scale (750K images, 7.5M QAs) and focus on CT/MRI fill a clear need in the community for large, diverse radiology VQA resources.

**Questions To Address In The Rebuttal:**

Can the authors provide more detail (or examples) on the question templates and answer phrasing? How many distinct templates were used per task, and how are synonyms or alternative expressions handled?

---

> ### Author Response · Authors · 2026-01-24
> **Response to Reviewer WnEX**
>
> We thank the reviewer for the supportive evaluation and for noting RadImageNet-VQA’s scale and diversity, along with the extensive VLM benchmarking. We clarify here the linguistic diversity of our templates and robustness to phrasing variation.
>
> Template-generated QA can indeed risk models exploiting bias. To mitigate bias, we define multiple template families per task (17 for Anatomy, 4 for Abnormality, and 10 for Pathology) and randomly sample a template for each instance, so each label appears under varied phrasings rather than a single fixed string. The text-only experiments (Section 4.2) support that this strategy mitigates linguistic priors in RadImageNet-VQA. Regarding answers, ground-truth answers follow expert-curated RadImageNet metadata (e.g., “bronchiectasis”). While this ensures strict clinical precision for evaluation, we acknowledge that real-world dialogue is more varied and future work could involve characterizing the severity or stage of pathologies (e.g., acute vs. chronic).
>
> In response to this comment, we have added the following to the revised manuscript:
> * an out-of-template experiment with manually edited, unseen questions (e.g., laterality “right/left”) in Appendix C.4, showing fine-tuned models generalize beyond template strings;
> * additional qualitative predictions across formats in Appendix C.6, with a subset moved to the main paper (Section 4.4), including failure cases.
>
> Note also that for open-ended questions, our LLM judge evaluates semantic equivalence, making the scoring more tolerant to synonyms and minor paraphrases.

---

### Official Review · Reviewer_cYNV · 2026-01-10

**Confidence:** 4
**Preliminary Rating:** 3
**Final Rating:** 5

**Summary:**

RadimageNet-VQA is an extensive dataset that includes a large training set and constructed a strong benchmark for radiologic visual question answering. Included in this dataset are 8 unique anatomical regions and 97 different pathologies as well as three areas of focus; abnormalities, anatomy and pathology, that have been broken down into various question formats. The RadimageNet-VQA has approximately 750k images, along with 7.5M QAs making it one of the largest Med-VQA dataset.

**Strengths:**

1. The development of substantial and comprehensive datasets in Radiologic VQA remains a challenge. This work has therefore provided an important contribution to both the research communities in medical imaging and vision-language, as they provide a large scale VQA dataset for Radiology.
2. Extensive evaluations of state-of-the-art VLMs have been performed, showing that current models are able to perform well on questions relating to anatomy and perform basic abnormality detection tasks.

**Weaknesses:**

1. In Figure 5, the performance of the different models are poor for pathology identification. What is the justification for this low accuracy? Also some qualitative results might be helpful.
2. The lack of CXR data in the dataset is surprising. Given that CXR is one of the most basic and fundamental imaging modalities, it would stand to reason that RadImageNet would include CXR data. Given, the future direction in medical image analysis is towards foundational models which includes more types of imaging, it is prudent to have CXRs in benchmarking dataset like this.
3. The evaluation benchmark is stated to have a stratified sample of 1,000 images with 9,000 individual QA pairs per image; however, it is not clear how the validity of the QA pair was established. As such, it is important to clarify whether or not the QA pairs were validated by an individual with expertise in radiology or another area of expertise related to the images.
4. While the paper states that the dataset includes “8 anatomic regions and 97 pathologies, which were generated through prompt based templates,” the use of prompt based templates has not been sufficiently described.

**Detailed Comments:**

Overall, the reviewer is satisfied with what this paper's contribution is but believes that addressing the weaknesses outlined above will increase the overall quality of this paper.

**Justification Of Final Rating:**

The authors have provided clear and satisfactory clarifications to all of my questions. I am broadly satisfied with both the paper and the rebuttal. Moreover, the paper proposes a dataset that is highly valuable to the medical imaging and deep learning community. I therefore recommend a `strong accept'.

**Justification Of The Preliminary Rating:**

The paper makes a strong contribution by introducing a large-scale VQA dataset, but concerns regarding pathology-level performance, absence of CXR, and insufficient methodological clarity justify a borderline assessment.

**Questions To Address In The Rebuttal:**

Please see the weakness section.

---

> ### Author Response · Authors · 2026-01-24
> **Response to Reviewer cYNV**
>
> We thank the reviewer for the constructive feedback and for recognizing RadImageNet-VQA as an important contribution to medical vision-language research, including its scale and the breadth of our benchmarking. We hope the following responses address your concerns.
>
> &nbsp;
>
> **[W1- Pathology identification performance]**
>
> The low accuracy in pathology identification is indeed a key finding of the benchmark. Identifying fine-grained pathologies in a free-form VQA setting requires recognizing subtle visual patterns within specific anatomical contexts. For instance, models reliably identify large or high-contrast abnormalities such as gross joint dislocations (ankle/foot osseous disruption) or large fluid collections, but they struggle with small, localized lesions like meniscal tears, chondral abnormalities, or marrow signal changes in the spine, where fine texture, shape, and anatomical context are crucial. To further analyze this "pathology bottleneck," we have expanded the revised manuscript to include qualitative failure cases across open-ended, closed-ended, and multiple-choice formats (Section 4.4 and Appendix C.5).
>
> &nbsp;
>
> **[W2 - Absence of chest X-ray data]**
>
> We agree that chest X-ray is an important modality, but it is already covered by existing large-scale resources like ReXVQA (600K+ QA pairs), as summarized in Table 1. In contrast, CT and MRI modalities remain comparatively underrepresented in medical VQA despite being central to complex clinical diagnostic tasks. Our CT/MRI focus is therefore a deliberate design choice to fill this gap, as recognized by reviewer WnEX. The presence of the two 3D modalities also enables potential cross-modality studies in future work.
>
> &nbsp;
>
> **[W3 - Validity of benchmark’s QA pairs]**
>
> The QA pairs are grounded directly in the expert-curated labels of the original RadImageNet test split and generated via a template-based pipeline that explicitly incorporates the ground-truth label into the question. The task-specific templates are randomly sampled to introduce linguistic variation. For multiple-choice questions, we include distractors to reduce trivial answer elimination (Section 3.1, p.6). The text-only experiments (Section 4.2) confirm that models cannot solve these questions through linguistic shortcuts alone.
>
> &nbsp;
>
> **[W4 - Prompt-based templates]**
>
> Our prompt-based templates are used to generate the question formulations; the 8 anatomical regions and 97 pathologies are taken directly from RadImageNet’s expert-curated labels (they are not generated by prompts). Regarding the reviewer’s question, we suspect the sentence cited from page 2 is ambiguous and we have rephrased it in the revised manuscript to clarify this point. We also refer the reviewer to Figure 7 (Appendix A.2), which lists all templates used in the dataset (17 for Anatomy, 4 for Abnormality, and 10 for Pathology). During generation, templates are randomly sampled to introduce linguistic variety while keeping the clinical labels precise.

---

### Author Rebuttal · Authors · 2026-01-24

**Rebuttal:**

We thank the reviewers for their constructive and supportive feedback. We appreciate the consensus that RadImageNet-VQA is a valuable contribution to medical VQA. We are encouraged by the emphasis on its scale (dPjo, WnEX, cYNV), diversity (WnEX), and the clinical relevance of our task decomposition (dPjo). We also value the positive assessment of the extensive evaluation across state-of-the-art VLMs (cYNV, WnEX, dPjo) and the additional analyses, including text-only ablations (dPjo). We revised the manuscript to improve clarity, strengthen validation, and incorporate reviewers’ comments.

Revisions in the updated manuscript are highlighted in blue and include:
* **Generalization analysis**: we add an out-of-template experiment with manually edited, unseen questions to evaluate generalization (Appendix C.4). We also add a template-stratified performance breakdown to quantify inter-template variability (Appendix C.3).
* **In-depth qualitative analysis** in Section 4.4 and Appendix C.5, comparing model predictions and including failure cases to understand why pathology identification remains challenging compared to anatomy recognition.
* **LLM-as-a-judge reliability**: we add a manual evaluation on 400 open-ended predictions with agreement analysis against the LLM judge (Section 3.2 and Appendix B.2).
* **Expanded discussion and limitation details**, including ethical considerations and potential dataset biases inherited from RadImageNet to improve transparency (Section 5).

**Supporting Material:**

/attachment/571a4ec95b6894f85c71f7ac0df2fb520811918e.pdf

---

### Meta-Review · Area_Chair_pxpt · 2026-02-09

**Recommendation:** Accept (Oral)
**Confidence:** 5

**Metareview:**

All reviewers provided positive ratings (2x **Strong Accept** + 1x **Week Accept**). The decision follows the reviewers’ final ratings. The authors are encouraged to thoroughly address the highlighted issues and integrate key findings from the rebuttal into the final version.

---

### Decision · Program_Chairs · 2026-02-14

Accept (Poster)